# Molecular and cellular immune features of aged patients with severe COVID-19 pneumonia

Domenico Lo Tartaro [1], Anita Neroni[1], Annamaria Paolini[1], Rebecca Borella[1], Marco Mattioli [1], Lucia Fidanza[1], Andrew Quong[2], Carlene Petes[2], Geneve Awong[2], Samuel Douglas[2], Dongxia Lin[2], Jordan Nieto[2], Licia Gozzi [3], Erica Franceschini[3], Stefano Busani[4,5], Milena Nasi[4], Anna Vittoria Mattioli[4,6], Tommaso Trenti[7], Marianna Meschiari [3], Giovanni Guaraldi [3,4], Massimo Girardis[4,5], Cristina Mussini[3,4], Lara Gibellini [1], Andrea Cossarizza [1,6,8✉] & Sara De Biasi [1,8✉]

Aging is a major risk factor for developing severe COVID-19, but few detailed data are available concerning immunological changes after infection in aged individuals. Here we describe main immune characteristics in 31 patients with severe SARS-CoV-2 infection who were >70 years old, compared to 33 subjects <60 years of age. Differences in plasma levels of 62 cytokines, landscape of peripheral blood mononuclear cells, T cell repertoire, transcriptome of central memory CD4$^+$ T cells, specific antibodies are reported along with features of lung macrophages. Elderly subjects have higher levels of pro-inflammatory cytokines, more circulating plasmablasts, reduced plasmatic level of anti-S and anti-RBD IgG3 antibodies, lower proportions of central memory CD4$^+$ T cells, more immature monocytes and CD56$^+$ pro-inflammatory monocytes, lower percentages of circulating follicular helper T cells (cTfh), antigen-specific cTfh cells with a less activated transcriptomic profile, lung resident activated macrophages that promote collagen deposition and fibrosis. Our study underlines the importance of inflammation in the response to SARS-CoV-2 and suggests that inflammaging, coupled with the inability to mount a proper anti-viral response, could exacerbate disease severity and the worst clinical outcome in old patients.

[1] Department of Medical and Surgical Sciences for Children and Adults, University of Modena and Reggio Emilia School of Medicine, Via Campi 287, 41125 Modena, Italy. [2] Fluidigm Corporation, 2 Tower Place, Suite 2000, South San Francisco 94080 CA, USA. [3] Infectious Diseases Clinics, AOU Policlinico di Modena, via del Pozzo 71, 41124 Modena, Italy. [4] Department of Surgery, Medicine, Dentistry and Morphological Sciences, University of Modena and Reggio Emilia, via del Pozzo 71, 41124 Modena, Italy. [5] Department of Anesthesia and Intensive Care, AOU Policlinico and University of Modena and Reggio Emilia, via del Pozzo 71, 41124 Modena, Italy. [6] National Institute for Cardiovascular Research, via Irnerio 48, 40126 Bologna, Italy. [7] Department of Laboratory Medicine and Pathology, Diagnostic Hematology and Clinical Genomics, AUSL/AOU Policlinico, 41124 Modena, Italy. [8] These authors jointly supervised this work: Andrea Cossarizza, Sara De Biasi. ✉email: andrea.cossarizza@unimore.it; debiasisara@yahoo.it

An overwhelming preponderance of cases of Coronavirus disease (COVID-19) and death has been observed within the elderly population, especially in persons with pre-existing conditions and co-morbidities[1]. Indeed, major risk factors for developing a severe COVID-19 disease include age, male, sex, obesity, smoking, and comorbid chronic conditions such as hypertension, cardiovascular diseases and type 2 diabetes[2].

Immunity is a cornerstone of host-pathogen interaction in any infectious disease, but its functionality decreases in elderly. Age-related remodelling of the immune system, i.e., immunosenescence, and the persistent, chronic and subclinical state of inflammation that has been defined "inflammaging" can predispose to a variety of infections[3,4], including those by viruses like SARS-CoV-2[5].

Noteworthy, first of all aging is associated with reduced expression of ACE2, and low expression of ACE2 after infection potentially mediates a pro-inflammatory state through the production of angiotensin (Ang) II[6]. Second, chronic activation of monocytes, a typical feature of inflammaging, predisposes aged individuals to create a pro-thrombotic environment, which further contributes to the hyperinflammatory, pathophysiological response that causes the devastating outcomes observed in severe COVID-19[7]. Third, the repertoire of T lymphocytes can change with age because of the expansion of some clones[8], and its restriction influences the accessibility of naïve T cells to SARS-CoV-2 antigens, resulting in a delayed priming and activation of specific cells.

The immune system of patients with severe COVID-19 is characterized by several features, such as, among others, cytokine storm, high levels of plasma d-dimer, lymphopenia and high neutrophil-lymphocyte ratio. More in details, they are characterized by alterations in the neutrophil and monocyte compartments, exhausted and senescent T cells, and high levels of circulating plasmablasts[9–15]. Even if a number of studies have described such alterations, no comprehensive investigation exists in patients aged more than 70 years.

Here, we describe relevant immune changes in 31 aged patients with severe SARS-CoV-2 infection (mean age: 76.4 years), compared to 33 adult patients at the same stage of infection (mean age: 49.8 years). Plasma levels of 62 molecules, including cytokines, chemokines and growth factors were measured, along with main clinical and biochemical parameters. By using 38-parameter mass cytometry, we investigated peripheral mononuclear cell (PBMC) populations, paying particular attention to the compartments of T lymphocytes, B cells and monocytes. In silico analyses were performed to identify further alterations of the immune response either in peripheral cells or in those resident in the lung. Besides being characterized by higher plasma levels of pro-inflammatory cytokines, a profound dysregulation exists in several immune compartments of aged SARS-CoV-2 patients. Such changes were more relevant within CD4+ central memory T cells and intermediate monocytes, revealing that in these individuals a more marked process of inflammaging and immunosenescence is present, and could be responsible, at least in part, for the more severe course of the infection that occurs with age.

## Results
**Patients.** We studied a total of 64 patients with severe COVID-19 pneumonia admitted into the Infectious Diseases Clinics or to Intensive Care Unit (ICU) of the University Hospital in Modena over the period of March 2020-February 2021. All patients had documented acute SARS-CoV-2 infection and displayed symptoms including fever, cough, fatigue, and were classified as severe on the basis of the WHO's scale[16].

Patients were divided into two groups: those aged <60 years (a total of 33 individuals, named "COVID-under", CUN) and those aged >70 years (a total of 31 patients, named "COVID-over", COV), as in Supplementary Data 1. Immunological features of CUN and COV are compared as described below. The clinical characteristics of CUN and COV patients are described in Table 1. Male represents 67% of COV 90% of CUN patients (Fisher's exact test, $p < 0.03$).For some analyses, patients were compared to a total of 32 healthy adults with a mean age of 54.1 ± 15.9years.

**COV patients showed higher plasma level of IL-6 and lower plasmatic levels of IFN-γ and wound healing growth factors if compared to CUN.** Plasma level of 62 cytokines in 26 CUN and 23 COV patients have been grouped according to one of their main functions, as reported in Fig. 1. COV patients showed increased plasma level of IL-6 and IL-11 (Fig. 1a). These two pro-inflammatory cytokines are involved in acute response to viruses and in lung fibrosis, respectively[17,18]. In COVID-19 patients, high levels of IL-6 have been associated with death[2], while during SARS-CoV-2 infection until now no alterations have been described in the levels of IL-11, a molecule which is linked to the hyperreactivity of the airways during viral infection.

Despite higher levels of IL-6, COV patients displayed lower levels of IL-1β, IL-1α, IL-2, IFN-β, IFN-γ and IL-33 if compared to CUN patients (Fig. 1a). These cytokines are involved in the regulation of inflammation, T cell activation, antiviral response, dendritic cell (DC) recruitment, tissue remodeling and hematopoietic progenitor cell mobilization[19,20]. Moreover, lower plasma levels of IFN-γ have been associated with increased lung fibrosis[21].

Regarding chemokines, we observed higher levels of fractalkine/CX3CL1 in COV if compared to CUN patients. This molecule is chemotactic for non-classical monocytes, and it is significantly increased compared to healthy controls in patients with severe COVID-19, independent of time after symptom onset[22]. COV patients showed also lower levels of molecules able to recruit neutrophils in the site of infection, such as CXCL2 and CCL5 (Fig. 1b). Soluble molecules involved in lung tissue repairing and vascular remodeling such as epidermal growth factor (EGF), platelets derived growth factor (PDGF-AA) and PDGF-AA/BB, were lower in COV if compared to CUN patients (Fig. 1c).

We measured other soluble molecules, such as FAS, FAS-L and PD-L1, involved in different apoptotic pathways[23]. Higher levels of soluble FAS were found in COV, as compared to CUN patients (Fig. 1d).

CUN and COV patients displayed similar plasma levels of the remaining 42 cytokines that we analysed (Supplementary Fig. 1). The hyper-inflamed status for both CUN and COV was also highlighted by the high levels of soluble GM-CSF, TNF, granzyme B and IL-18 as compared to healthy donors (HD) (Supplementary Fig. 1).

**Different distribution and proliferative capacity of T and B cell subsets in CUN and COV patients.** To deeply characterize the landscape of human PBMC of during SARS-CoV-2 infection, we interrogated a total of 17 severe COVID-19 patients (CUN = 7, COV = 10). Peripheral blood mononuclear cells (PBMC) were stained and analyzed using a 38-markers mass cytometry panel. Unsupervised analysis revealed 23 different clusters, representing myeloid and lymphoid compartments (Fig. 2a, b).

Among CD4+ T cells, we identified three main cell populations, i.e., naïve (CD4+CCR7+CD45RA+CD45R0−CD27+ CD28+), central memory [CD4+CCR7+CD45RA− CD45R0+ CD27+CD28+ (CM)] and effector memory [CD4+CCR7− CD45RA− CD45R0+CD27−CD28− (EM)]. CD8+ T cells were classified in four clusters, i.e., naïve [CD8+CCR7+CD45RA+

**Table 1 Demographic and clinical characteristics of COVID-19 patients.**

| Variable | COVID UNDER (n = 33) | COVID OVER (n = 31) | *p*-value |
|---|---|---|---|
| Demographic characteristics | | | |
| Age (mean years, range)[a] | 49.8 (34 – 59) | 76.4 (69 – 88) | <0.0001 |
| Sex (M, %)[b] | 67.0 | 90.3 | 0.03 |
| Deceased, N (%)[b] | 1 (3.0) | 16 (51.6) | <0.0001 |
| Clinical characteristics | | | |
| Respiratory rate (±SD)[a] | 23.5 (±6.6) | 25.6 (±6.6) | ns |
| Heart rate (±SD)[a] | 82.2 (±11.5) | 82.2 (±22.0) | ns |
| SOFA score, mean (range)[a] | 2.32 (1 – 5) | 3.91 (1 – 6) | 0.0006 |
| Coexisting conditions | | | |
| Type 2 diabetes, N (%)[b] | 6 (18.2) | 10 (32.2) | ns |
| Cardiovascular Dis., N (%)[b] | 7 (21.2) | 15 (48.4) | ns |
| Chronic Kidney Dis., N (%)[b] | 2 (6.1) | 7 (25.8) | 0.044 |
| Cancer., N (%)[b] | 1 (3.0) | 1 (3.2) | ns |
| Arterial blood gas analysis | | | |
| $pO_2$, mmHg (±SD)[a] | 73.0 (±26.8) | 61.99 (±10.1) | ns |
| $sO_2$, % (±SD)[a] | 93.6 (±3.9) | 91.8 (±3.3) | ns |
| $pCO_2$, mmHg (±SD)[a] | 37.6 (±4.7) | 35.6 (±5.9) | ns |
| $pO_2/FiO_2$ (±SD)[a] | 149.9 (±80.2) | 151.8 (±86.2) | ns |
| Clinical blood parameters | | | |
| ALT, U/L (±SD)[a] | 39.3 (± 20.5) | 34.41 (±21.38) | ns |
| Total bilirubin, mg/dL (±SD)[a] | 0.6 (±0.2) | 0.8 (±0.4) | ns |
| CK, U/L (±SD)[a] | 125.2 (±121.3) | 166.6 (±138.3) | ns |
| Creatinine, mg/dL (±SD)[a] | 0.8 (±0.2) | 1.68 (±1.4) | <0.0001 |
| D-dimer, ng/mL (±SD)[a] | 1221.8 (±542.2) | 1612.86 (±1012.48) | ns |
| LDH, U/L (±SD)[a] | 673.4 (±203.1) | 677.3 (±232.3) | ns |
| CRP, mg/dL (±SD)[a] | 12.2 (±8.3) | 11.68 (±7.58) | ns |
| Blood cell count | | | |
| White blood cells, N/uL (±SD)[a] | 6369.8 (±2419.5) | 6422.8 (±2057.0) | ns |
| Lymphocytes, N/uL (±SD)[a] | 1028.7 (±405.4) | 898.1 (±347.84) | ns |
| Neutrophils, N/uL (±SD)[a] | 5056.9 (±3488.7) | 4898.97 (±1888.9) | ns |
| Platelets, N*$10^9$/L (±SD)[a] | 261.8 (±139.9) | 199.45 (±106.24) | 0.008 |

*SD* standard deviation, *Dis.* disease, *ns* non-significant value, *M* male, *pCO₂* partial pressure of carbon dioxide, *pO₂* partial pressure of oxygen, *sO₂* oxygen saturation, *pO₂/FiO₂* fraction of inspired oxygen, *ALT* alanine aminotransferase, *CK* creatine kinase, *LDH* lactate dehydrogenase, *CRP* C-reactive protein.
[a]Unpaired *t* test or Mann-Whitney *U* test.
[b]Fisher's exact test.

CD45R0⁻CD27⁺CD28⁺], activated effector memory [CD8⁺ CCR7 − CD45RA⁻CD45R0⁺CD27⁺CD28⁺HLA-DR⁺CD38⁺ (act EM)] and two clusters of effector memory cells re-expressing CD45RA [CD8⁺CCR7⁻CD45RA⁺CD45RO⁻CD27⁻CD28⁻ (EMRA)] expressing or not CD57. We also found double positive T cells [CD3⁺CD4⁺CD8⁺ (DP)] and two clusters of unconventional T lymphocytes, i.e., mucosal associated invariant T cells [CD3⁺CD8⁺CD161⁺ (MAIT)] and gamma-delta T cells (CD3⁺TCRγδ⁺).

B lymphocytes were identified on the basis of the expression of CD19, and were classified in four clusters, i.e., naïve (CD19⁺CD20⁺IgD⁺IgM⁺CD21⁺CD24⁺CD40⁺CXCR5⁺), memory (CD19⁺CD20⁺IgD⁻IgM⁻CD21⁺CD24⁺CD40⁺CXCR5⁺), plasmablasts (CD19lowCD20⁻ IgD⁻IgM⁻CD80⁺CD38⁺) and atypical cells (CD19⁺CD20⁺CD40⁺HLA-DR⁺CCR6lowCD1-clowCD27⁻CD21⁻CD24⁻CD38⁻IgD⁻IgM⁻)[24]. Three populations of natural killer (NK) cells were identified according to the expression of CD56 and CD16: those defined as early NK (CD16lowCD56⁺CD57⁻) and those mature (CD16highCD56⁺), expressing or not CD57. CD57⁺ NK cells exhibit both memory-like features and potent effector functions and are one of the hallmarks of ageing[25].

Regarding myeloid compartment, monocytes were identified as CD14⁺CD11b⁺CD11c⁺HLA-DR⁺ cells (Fig. 2b). DC were identified according to the expression of CD123. Four populations of DC were recognized: two populations of plasmacytoid DC [CD123⁺CD11c⁻ (pDC)] expressing or not CXCR3, and two of myeloid DC [CD123⁺CD11c⁺ (mDC)], one of which is activated

expressing both CD38 and HLA-DR. Finally, we identified low density neutrophils [CD66b⁺CXCR1⁺CCR6⁺ (LDN)] as those cells present among mononuclear cells after isolation by using Ficoll Hypaque, as previously described[26] (Fig. 2b).

COV patients displayed a strong reduction of CM CD4⁺ T cells and naïve CD8⁺ T cells ($p = 0.00036$ and $p = 0.00036$, respectively, Fig. 2c). Moreover, COV showed higher percentages of mature NK CD57 + cells if compared to CUN patients.

Among CD19⁺ cells, percentages of atypical B cells were similar in both COV and CUN patients, while the percentages of memory B cells were significantly lower in COV patients ($p = 0.021$; Fig. 2c). The percentage of plasmablasts was higher in COV if compared to CUN patients ($p = 0.044$, Fig. 2c). Similar percentages of all other subpopulations were found in CUN and COV patients. Reclustering of monocytes and of CD4⁺ and CD8⁺ T cells was performed to better describe the populations with a deeper resolution and described below.

The proliferative capacity of T and B cells was assessed along with phenotypic analysis, and we found that CM CD4⁺ T cells from COV showed a reduction of both proliferation index (PI) and percentage of divided cells (PD) if compared to CUN patients (Fig. 2d). This defective proliferative capacity could partially explain the reduction of this population in the circulation of COV patients. No differences were found within the CD8⁺ T cell subset (Supplementary Fig. 2). B cells from COV displayed higher PI and PD if compared to CUN patients, which likely explains the higher percentages of circulating plasmablasts observed in COV patients.

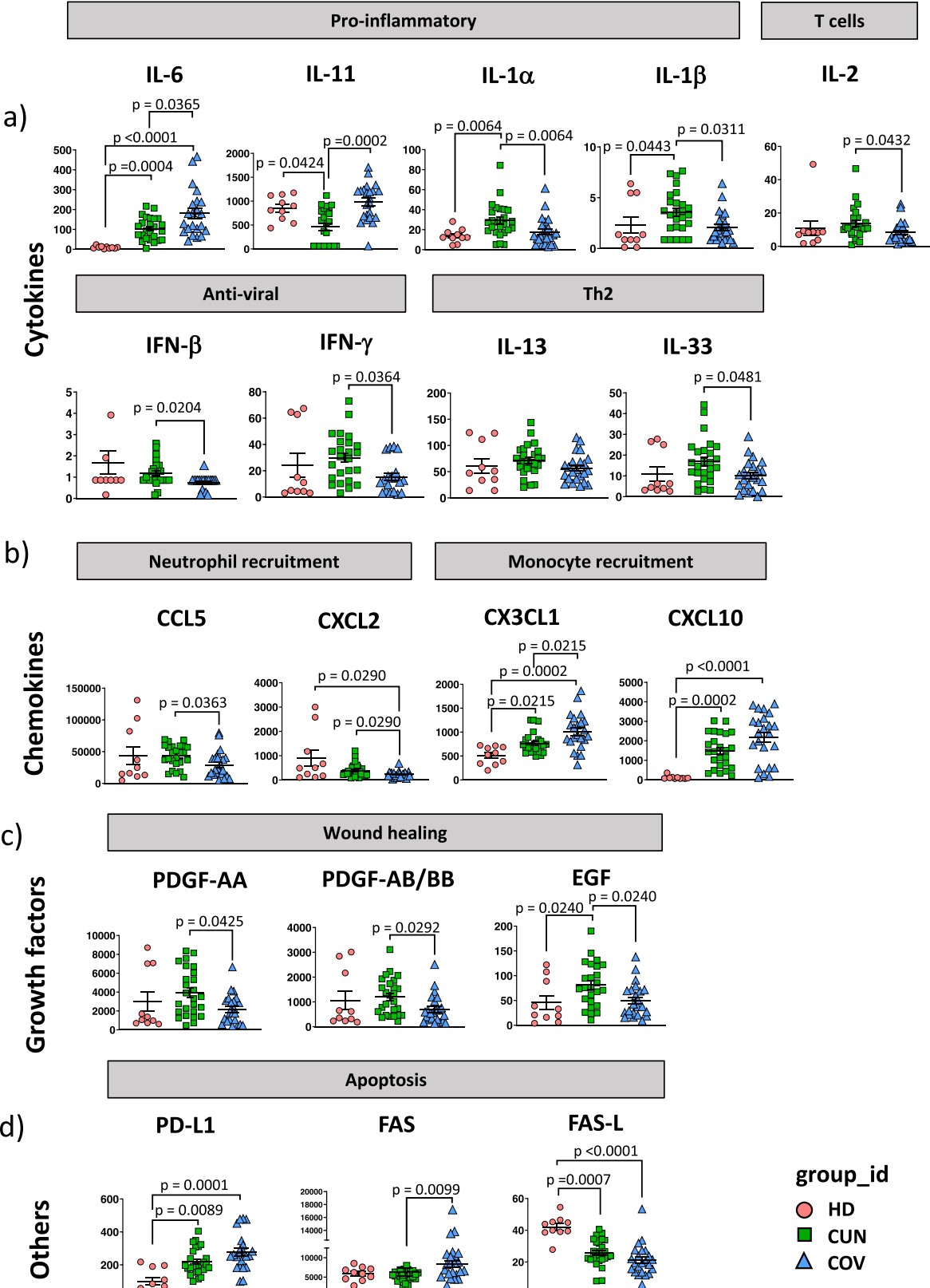

**Fig. 1 Plasma level of cytokines, chemokines and growth factor from young (CUN) and aged (COV) patients with severe COVID-19.** Scatter plots of 19 cytokines and other mediators in plasma obtained from healthy donors (HD, *n* = 10), CUN (*n* = 26) and COV (*n* = 23) subjects. Scatter plots show individual values, mean ± SEM. Kruskal-Wallis test with Benjamini-Hochberg correction for multiple comparisons was used to test the differences among the three groups; *p*-values are indicated in the figure. The upper bar indicates the function mediated by each plasmatic molecule. **a** Dotplots show the level of IL-6, IL-11, IL-1α, IL-1β, IL-2, IFN-β, IFN-γ, IL-13 and IL-33. **b** Dotplots show the level of CCL5, CXCL2, CX3CL1 and CXCL10. **c** Dotplots show the level of PDGF-AA, PDGF-AB/BB and EGF. **d** Dotplots show the level of PD-L1, FAS and FAS-L.

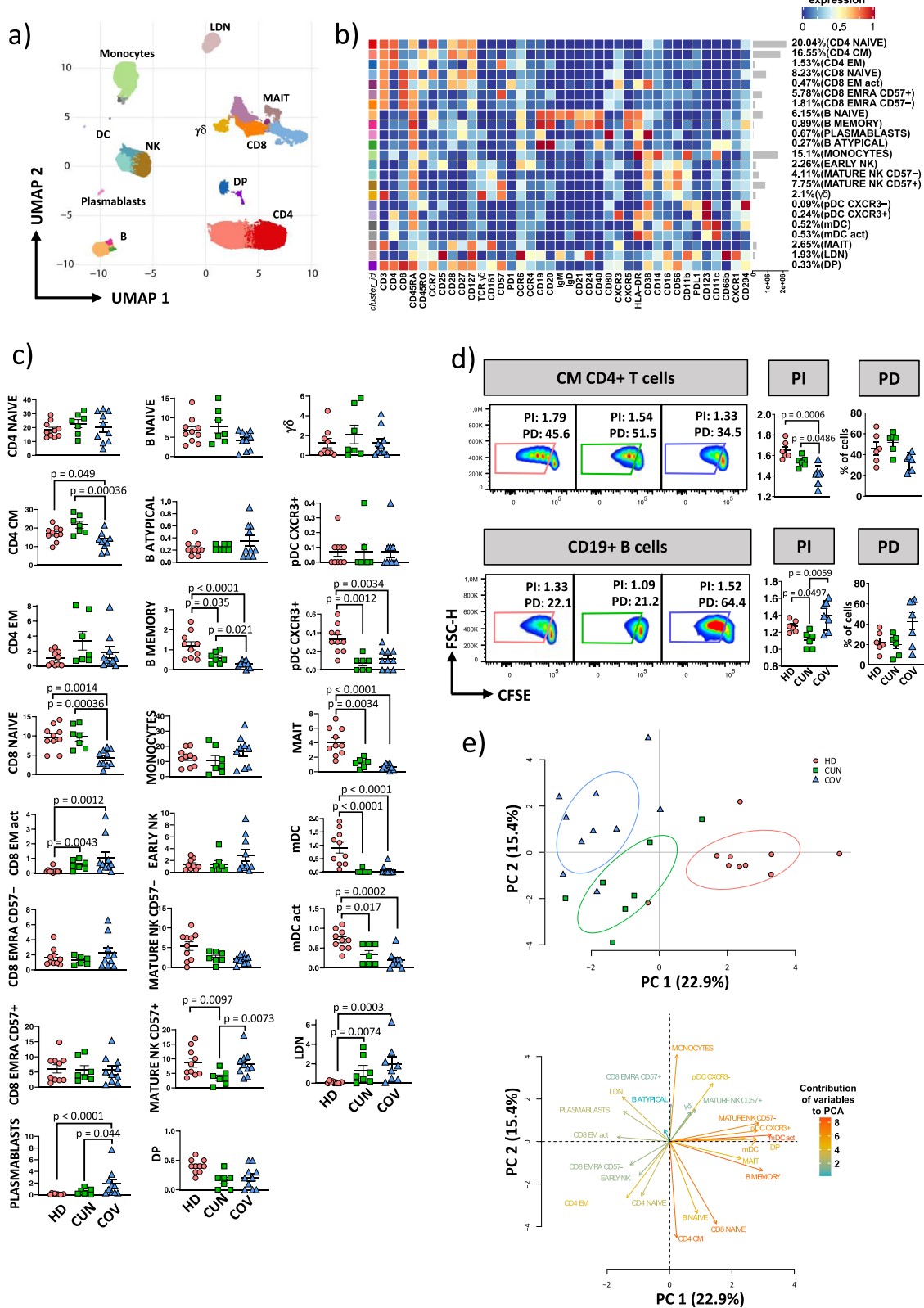

The principal components analysis (PCA), related to data regarding the complete phenotype obtained by CyTOF, revealed that COV and CUN patients cluster in different positions of the two-dimensional PCA space (Fig. 2e, upper panel). Immune features related to the amount of LDN, plasmablasts and activated EM

CD8[+] T cells (more abundant in COV patients) were the main drivers of the clusterization of patients in two different areas (Fig. 2e, lower panel). The same panel indicates that CUN patients were characterized by elevated frequencies of CM CD4[+] T cells, naïve CD4[+] T cells, naïve CD8[+] T cells, naïve B cells and early NK cells.

**Fig. 2 Deep immune profiling of PBMC from severe COVID-19 patients. a** UMAP plot shows the 2D spatial distribution of 11,153,288 cells from healthy donors (HD, $n = 10$), CUN ($n = 7$) and COV ($n = 10$) embedded with FlowSOM clusters. **b** Heatmap of the median marker intensities of the 38 lineage markers across the 23 cell populations obtained with FlowSOM algoritm after the manual metaclusters merging. The colors of cluster_id column correspond to the colors used to label the UMAP plot clusters. The color in the heatmap is referred to the median of the *arcsinh* marker expression (0 to 1 scaled) calculated over cells from all the samples. Blue represents lower expression while red represent higher expression. Light grey bar along the rows (clusters) and values in brackets indicate the relative sizes of clusters. CM central memory, TM transitional memory, EM effector memory, EMRA effector memory re-expressing the CD45RA, NK natural killer cells, γδ T cells expressing the γδ T cell receptor, mDC myeloid dendritic cells; pDC plasmacytoid dendritic cells; MAIT mucosal associated invariant T cells; LDN low density neutrophils; DP double positive CD4 + CD8 + T lymphocytes. **c** Dotplots show the relative cells percentage of the 23 clusters among healthy donors (HD; salmon circles; $n = 10$), CUN (green squares; $n = 7$) and COV patients (blue triangles; $n = 10$). The central bar represents the mean ± SEM. The statistical relevant adjusted *p*-values obtained by GLMM statistical test comparing CUN and COV cluster percentages are reported in the figure. **d** Carboxyfluorescein diacetate succinimidyl ester (CFSE) dilution in CM CD4$^+$ T and CD19$^+$ B cells of HD (salmon), CUN (green) and COV (blue) samples after 16 hours of stimulation with 1 µg/mL of anti-CD3 plus anti-CD28. (Left) Dotplots show representative CFSE dilution in HDs, CUN and COV samples; (Right) paired dotplots show the proliferation index (PI) and percentage of divided cells (PD) of HD (salmon, $n = 6$), CUN (green, $n = 6$) and COV (blue, $n = 7$) samples after stimulation. Kruskal-Wallis test with Benjamini-Hochberg correction for multiple comparisons was used to test the differences among the three groups; *p*-values are indicated in the figure. The central bar represents the mean ± SEM. **e** (Top) Principal component analysis (PCA) using the percentage of clusters from HDs, CUN and COV subjects obtained by unsupervised analysis of PBMCs (see Fig. 2a). HD, salmon circles ($n = 10$); CUN, green squares ($n = 7$) and COV, blue triangles ($n = 10$); (Bottom) Contribution of the different variables to PCA. The color of the arrows underlines the contribution level, while the position underlines the positive or negative contribution. Negatively correlated variables are positioned on opposite sides of the plot origin (opposed quadrants).

**CM CD4$^+$ T cells in COV are transcriptionally different from those of CUN patients**. Given that in COV the percentage of CM CD4$^+$ T cells was lower and that such cells were characterized by lower proliferative potential than those from CUN patients, we isolated this cell subset and analyzed their transcriptome by RNA-sequencing. We could study 3 CUN and COV patients (median age $46.3 ± 8.1$ and $73.3 ± 5.9$ years, respectively), who have been matched with 3 young (HUN) and 3 aged (HOV) healthy subjects (median age $50.7 ± 4.9$ and $75.0 ± 6.4$ years, respectively).

By analysing the differentially expressed genes (DEGs), we observed that a set of 21 immune-related genes clearly separates infected patients from healthy subjects (Fig. 3a). Genes like *DUSP4*, *NR4A1*, *TBX21*(T-bet), *ZEB2*, *CEBPA*, *SIGLEC5* and *CIITA*, involved in Th1 priming and T-cell receptor (TCR) response[27], were expressed at higher level in HUN and HOV if compared to CUN and COV patients. On the contrary, CM T cells from COV and CUN patients were clearly distinct from those of healthy donors for higher levels of *CLTA4*, *LAG3*, *MCM6*, *MKI67* and *IFI27* genes, that are associated with T cell exhaustion, proliferation, and antiviral activity. Both CUN and COV groups, compared with their respective control group, expressed higher level of genes involved in the antiviral response and proliferation, like *IFIT5*, *IFI27* and *PIM1*, suggesting that those cells could have been activated by SARS-CoV-2 infection[28,29] (Supplementary Fig. 3b, c). When compared to their relative controls, COV, but not CUN patients expressed long non-coding RNAs like $SNTG2 − AS1$, $RGMB − AS1$, $ZNF32 − AS2$, $SH3BP5 − AS1$, and LAG3, well-known markers of age-related dysfunction and exhaustion, respectively[30,31] (Supplementary Fig. 3c). Several other DEGs were identified and are reported in the Supplementary Data 2.

Moreover, comparing CUN and COV patients, we found 7 differentially expressed genes (FDR < 0.05, Fig. 3b); 6 out 7 (*CTLA4*, *LAG-3*, *DUSP4*, *CXCR3*, *CCR5* and *LRRC32*) were upregulated in COV patients; 3 out 6 (*CTLA4*, *LAG-3*, *DUSP4*) were associated with T cells exhaustion and defective TCR response[31–33]. A different expression of these genes was not observed comparing HUN and HOV samples (Supplementary Fig. 3a).

To further investigate the putative exhaustion of central memory compartment of COV patients, we quantified by flow cytometry the expression of PD-1 in CD4 + CM T cells, and found that the amount of this molecule per cell was significantly higher in COV patients (Fig. 3c).

**Reclustering of CD4$^+$ T cells identifies different subpopulations of CM T cells, including circulating follicular T lymphocytes.** CM T cell subset is a heterogeneous population composed by elements that can migrate into the lymph node and towards the inflamed tissue thanks to the presence of different chemokine receptors, such as CXCR3[34]. To gain more insights into CM compartment, we reclustered CD4$^+$ T cells and identified 22 clusters. Besides naïve T cells, defined as CD45RA$^+$ CD27$^+$CD28$^+$CCR7$^+$, we found 7 clusters of CM T cells defined as follow: CM (CD45RA$^−$CD27$^+$CD28$^+$CCR7$^+$), activated CM [CD45RA$^−$CD27$^+$CD28$^+$CCR7$^+$CD38$^+$ (act CM)], CM Th1/Th2 [CD45RA$^−$CD27$^+$CD28$^+$CCR7$^+$CXCR3$^+$CCR4$^{low}$ CCR6$^{low}$ (CM Th1_2)], CM Th2 [CD45RA$^−$CD27$^−$CD28$^+$ CCR7$^+$CXCR3$^−$CCR4$^+$ (CM Th2)], CM PD-1 [CD45RA$^−$ CD27$^+$CD28$^+$CCR7$^+$CD38$^+$PD-1$^+$ (CM PD-1)], CM CXCR5 [CD45RA$^−$CD27$^+$CD28$^+$CCR7$^+$CXCR3$^−$CXCR5$^+$PD-1$^−$ (CM CXCR5)], and circulating follicular T cells [CD45RA$^−$CD27$^+$ CD28$^+$CCR7$^+$CXCR3$^+$CXCR5$^+$PD-1$^+$ (cTfh)]. Tfh cells that circulate in the blood have been identified as counterparts of germinal center Tfh. In particular, cTfh cells expressing CXCR3 are important in the response to influenza vaccine, inducing a strong antigen-specific antibody response[35].

Among TM, we found these 6 clusters, defined classical TM [CD45RA$^−$CD27$^−$CD28$^+$CCR7$^−$], TM Th1 expressing PD-1 [CD45RA$^−$CD27$^+$CD28$^+$CCR7$^−$PD-1$^+$CXCR3$^+$ (TM PD-1 Th1)], activated TM Th1 expressing PD-1[CD45RA$^−$CD27$^+$ CD28$^+$CCR7$^−$PD-1$^+$CXCR3$^+$CD38$^+$HLA-DR$^+$ (TM PD-1 Th1_act)], TM Th2 [CD45RA$^−$CD27$^−$CD28$^+$CCR7$^−$CCR4$^+$], TM Th2 expressing PD-1[CD45RA$^−$CD27$^−$CD28$^+$CCR7$^−$ CCR4$^+$ (TM PD-1 Th2)] and TM expressing both CD57 and PD-1 [CD45RA$^−$CD27$^+$CD28$^+$CCR7$^−$CD57$^+$PD-1$^+$CXCR3$^+$ (TM CD57 PD-1)]. We also identified two populations of EM, i.e., those CD45RA$^−$CD27$^−$CD28$^−$CCR7$^−$ and those expressing CD57 and PD-1 [CD45RA$^−$CD27$^−$CD28$^−$CCR7$^−$CD57$^+$PD-1$^+$ (EM CD57 PD-1)].

On the other side, three populations of effector memory cells re-expressing CD45RA (EMRA) were detected, i.e., [CD45RA$^+$ CD27$^−$CD28$^−$CCR7$^−$ (EMRA)], EMRA expressing CD57 [CD45RA$^+$CD27$^−$CD28$^−$CCR7$^−$CD57$^+$ (EMRA CD57)] and EMRA Th1/Th2 expressing CD57 [CD45RA$^+$CD27$^−$CD28$^−$ CCR7$^−$CD57$^+$CXCR3$^+$CCR4$^+$CCR6$^+$ (EMRA CD57 Th1_2)]. Finally, we identified also putative effector T regulatory

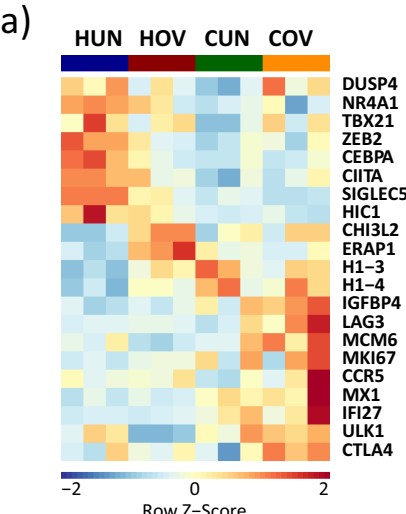

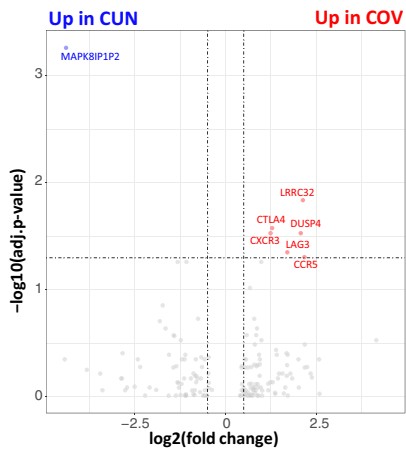

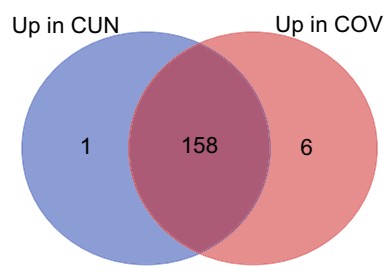

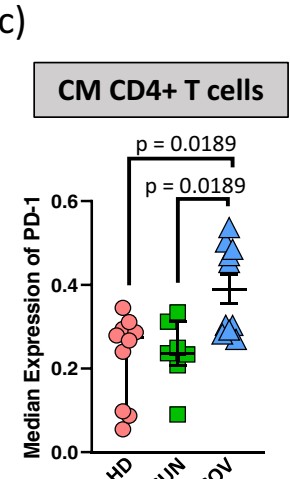

cells (eTreg) expressing CD45RA⁻CD127⁻CD25⁺CCR7⁻CCR4⁺ (Fig. 4a, b).

Besides different percentages of activated and effector memory subset of both CUN and COV patients compared to HDs (Fig. 4c), we found a higher percentage of activated CM in COV, but a lower percentage of cTfh cells if compared to CUN patients

(Fig. 4c, and Supplementary Fig. 4a, b). Moreover, we observed a negative correlation among cTfh and plasmablasts in COV patients, in contrast with that observed in CUN patients (Supplementary Fig. 4c).

To explore more in depth the correlation between cTfh and plasmablasts, we investigated in silico the plasma level of anti-

**Fig. 3 Transcriptomic analysis of sorted central memory (CM) CD4+ T cells. a** Heatmap showing the selection of 21 immune-related differential expressed genes (DEGs) in sorted CM CD4+ T of young healthy subjects (HUN, n = 3), aged healthy subjects (HOV, n = 3), CUN (n = 3) and COV (n = 3). Z-scores were calculated for each row (each gene) and used for the graphical visualization. **b** Left panel: volcano plot shows the DEGs among CM CD4+ T cells from COV and CUN patients. The genes upregulated in these cells from COV are in red, while those upregulated in CM CD4+ T cells from CUN in blue (thresholds: FDR < 0.05 and log₂FC > 0.5). Right panel: Venn diagram shows data summary of DEGs between CUN (blue circle) and COV (red circle). **c** Dotplot shows the surface median expression (number of molecules) of the PD-1 on CM CD4+ T cells from CUN and COV patients. The CM CD4+ T lymphocytes were selected from CyTOF unsupervised analysis (shown in Fig. 2). The central bar represents the mean ± SEM. Dashed line represents the median expression of the PD-1 on CM CD4+ T cells from HDs. Kruskal-Wallis test with Benjamini-Hochberg correction for multiple comparisons was used to test the differences among the three groups; p-values are indicated in the figure.

spike immunoglobulin (Ig) A1, A2, G1, G3, Fcγ receptor binding and Fc effector activity in both CUN and COV patients using publicly available data[36]. We selected 21 CUN and 18 COV with a median age of 49.0 ± 6.1 and 83.0 ± 5.9 years, respectively. We observed a reduced level of anti-S IgG3 isotype during the first and second week of infection in COV patients if compared to CUN, while the level of anti-RBD IgG3 isotype was lower in COV only during or second week of infection (Supplementary Fig. 5). Similar level of anti-S and anti-RBD IgA1, IgA2 and IgG1 were found. Given the differences in IgG class switching, we also examined the ability of SARS-CoV-2-specific antibodies to bind to the low-affinity Fcγ receptors. We observed that the IgG of COV patients were less able to bound both activating receptor FcγR2A (also known as CD32A) and FcγR3A (also known as CD16) during the first and second week of onset (Supplementary Fig. 5).

Finally, we reclustered CD8+ T cells, and found significant differences between patients and healthy donors. On the contrary, we could not identify any relevant difference between CUN and COV patients (Supplementary Fig. 6).

**SARS-CoV-2-reactive CD4+ cTfh cells from COV patients are more activated than those from CUN patients**. We performed an in silico analysis of published data containing single-cell transcriptomic and T cell receptor (TCR) analysis of >100,000 SARS-CoV-2-reactive CD4+ T cells[37]. In the original work, Meckiff et al. showed that SARS-CoV-2-reactive CD4 + T cells from hospitalized patients compared to non-hospitalized patients had increased cytotoxic follicular helper and classical T helper cells, but lower level of regulatory T cells. Although hospitalized patients exhibited different age (range 33–82 years), no differences were reported in terms of clusters percentage, gene expression and TCR clonality. From the entire dataset, we selected 8 severe patients that we could classified as CUN or COV on the basis of their age. Among 8378 SARS-CoV-2 specific CD4+ T cells, we found five clusters (Fig. 5a). Clusters 0 and 3 were characterized by high levels of *PRF1*, *GZMB*, *GZMH*, *GNLY*, and *NKG7* gene expression, and they were defined as cytotoxic CD4+ T cells (CTL) (Fig. 5b). However, given the different expression of *IFNG*, *TNF*, *CCL3* and *CCL4*, within these clusters we found CTL that were IFN^high and CTL IFN^low. Cluster 1 expressed high levels of *CD200*, *BTLA* and *POU2AF1* and was defined as formed by circulating follicular T helper (cTfh)[38]. Cluster 2 expressed high level of *STAT1*, *IL7R*, *SELL*, *TNFSF4* (also known as *OX40L*) characteristic of memory T cells that have recently engaged the Ag, and was defined as formed by activated-STAT1 + (act-STAT1) cells. Finally, cluster 4 displayed a transcriptional profile of proliferating cells expressing *MKI67*, *TOP2A*, *HMGB1-2* and *STMN1* (Fig. 5b).

Although no differences were found in the proportion of clusters between CUN and COV patients (Fig. 5c), the detailed analysis of their transcriptome revealed that CTL-IFN^high, CTL-IFN^low and act-STAT1 clusters from COV patients showed higher levels of *TIGIT*, *CTLA4*, *ICOS*, *HAVCR2* (TIM-3) and

*ZBED2* genes (Fig. 5d). In addition, the clusters from CUN patients displayed a transcriptional profile consistent with an increased antiviral activity, as they were expressing higher levels of genes related to IFN-response (*IFI27*, *IFI6*, *ISG15*), TNF-response (*TNF*, *TNFSF10*), cytotoxicity (*GNLY*), chemotaxis (*CXCR3*) and activation (*CD69*, *CD28*) (Fig. 5d). cTfh cluster of COV patients expresses fewer genes involved in T cells activation and monocytes maturation, such as *CD28, IFI27, CD74, CSF2* (also known as *GM-CSF*).

The analysis of TCR clonality was studied by detecting gene expression of SARS-CoV-2-specific CD4+ T cells, and revealed that COV patients were characterized by a lower quantity of small clonotypes within cTfh and activated (act)-STAT1 clusters (p < 0.0001; Fig. 5e). This suggests the possible existence of a depletion of clonal repertoire of the SARS-CoV-2-reactive CD4+ T cells in COV patients. In particular, despite the frequency of cTfh cell was similar in CUN and COV patients as reported above, cTfh of CUN were likely able to recognize a wider range of SARS-CoV-2 antigens.

Finally, we observed that a massive clonal expansion occurred within CD4-CTL IFN^high and CTL IFN^low clusters, in both CUN and COV groups, suggesting a probable active role of these cells in controlling the infection (Fig. 5e). Specifically, we observed that CTL IFN^high cells from CUN had a stronger expansion if compared to those from COV patients, as revealed by the higher level of hyperexpanded clones.

**Detailed monocyte's landscape revealed higher percentage of PD-L1+ intermediate monocytes in COV patients**. Reclustering of monocytes using markers such as CD14, CD16, CD4, CD56, CD38, CCR4, CCR6, CXCR3, CD294, CD80, CD40, CD11c, CD11b and PD-L1 allowed us to better characterize their different subpopulations and to point out differences between CUN and COV patients. Unsupervised analysis revealed 8 different clusters (Fig. 6a, b). Two of them were classified as classical (CL) monocytes, expressing or not CD56, two were intermediate (INT) monocytes expressing or not PD-L1, three were non-classical (NC) monocytes expressing or not CD38, CD40 or PD-L1, and one was classified as formed by immature cells (CD14^low). This detailed clusterization revealed the presence of one cluster of CL expressing CD56, who had been described as dysregulated monocyte producing basal level of IL-6 without stimulation[39], and that was more represented in COV patients (Fig. 6c).

Another peculiar cluster was that of NC monocytes expressing CD40 and PD-L1, observed predominantly in COV patients. These could be suppressive antigen-presenting cells involved in the control of the activation of the adaptive immune response. Here we observe that their frequency was higher in COV patients (Fig. 6c). Finally, we report that both CUN and COV patients display a significant percentage of immature monocytes, whose metabolism a particular population has been characterized recently[13]. Such cells express high level of CXCR3, a chemokine receptor that is able to recruit monocytes in the inflamed tissue,

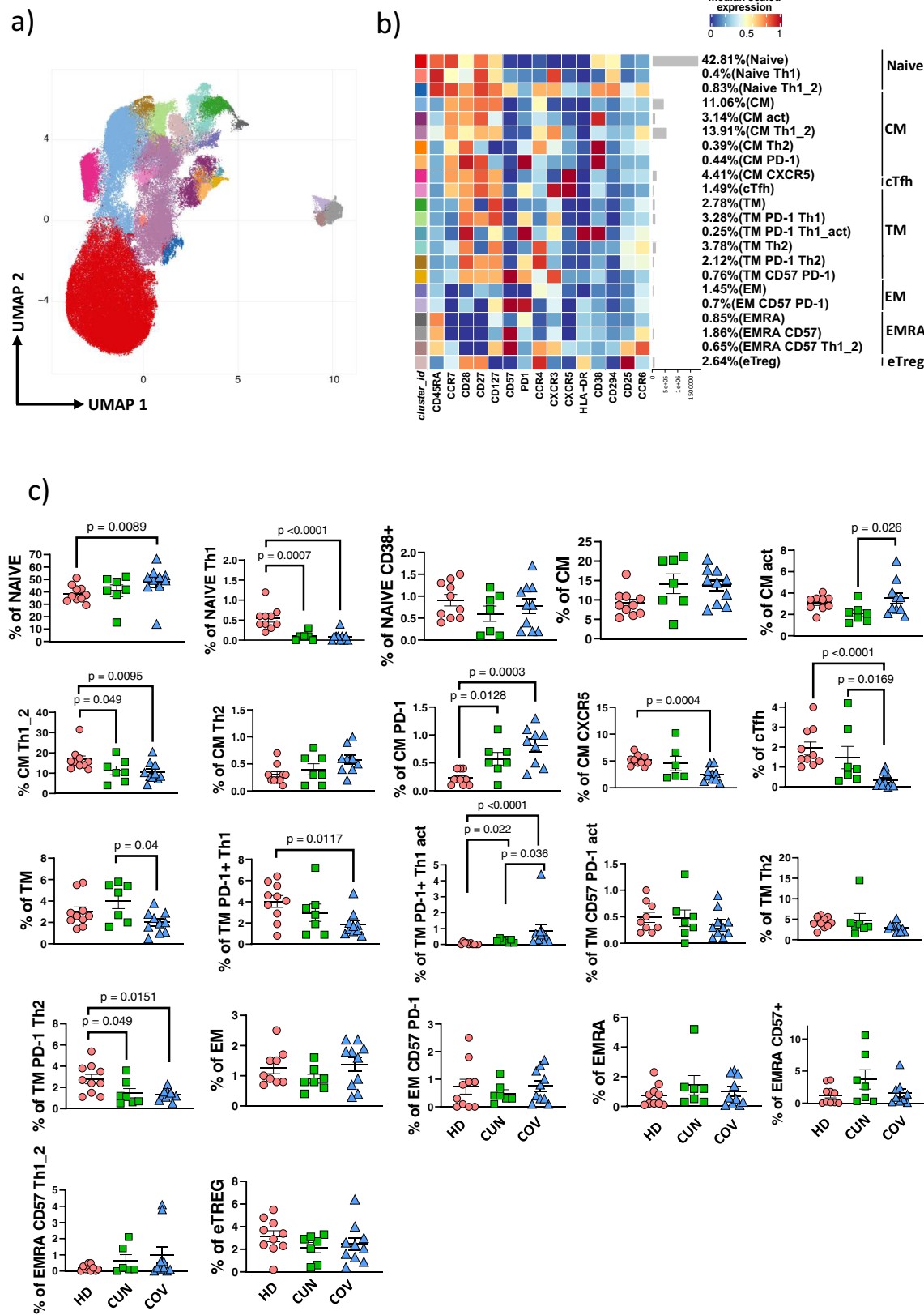

and thus could likely play a role in guiding the inflammation in the lungs. Furthermore, we showed that the expression of HLA-DR on total monocytes were lower in COV patients if compared to CUN patients (Fig. 6d), and this difference was due to a lower HLA-DR expression on different clusters, i.e., CL CD56⁻, CL CD56⁺, INT PD-L1⁺ (Supplementary Fig. 7a). In addition, all

monocytes from CUN and COV patients showed an upregulation of PD-L1 (Fig. 6d, Supplementary Fig. 7b).

Inflammatory molecules, including IL-6, inhibit HLA-DR and induce PD-L1 expression, suggesting that the high level of plasmatic IL-6 observed in COV patients may exacerbate the immune dysfunction of these patients. The amount of PD-L1 and

**Fig. 4 CD4+ T cells reclustering. a** UMAP plot shows the 2D spatial distribution of 4,252,208 cells from healthy subjects (HD, n = 0), CUN (n = 7) and COV (n = 10) embedded with FlowSOM clusters. **b** CD4+ T cells heatmap shows the median marker intensities of the 15 lineage markers across the 22 cell populations obtained with FlowSOM after the manual metaclusters merging. The color in the heatmap is referred to the median of the arcsinh marker expression (0 to 1 scaled) calculated over cells from all the samples. Blue represents lower expression while red represent higher expression. Lightgrey bar along the rows (clusters) and values in brackets indicate the relative sizes of clusters. CM, central memory; TM, transitional memory; EM, effector memory; EMRA, effector memory re-expressing the CD45RA; eTreg, effector T regulatory CD4+ T cells; cTfh, circulating T follicular helper cells. The black bar on the right is used to group subpopulations with similar immunophenotype backbone. **c** Dotplots show the relative cell percentage of the 22 clusters among healthy donors (HD; salmon circles; n = 10), CUN (green squares; n = 7) and COV (azure triangles; n = 10). The central bar represents the mean ± SEM. GLMM test was used for the statistical analysis. Exact p-values are reported in the figure.

HLA-DR molecules present on monocytes' surface was quantified by flow cytometry, and we found that both COV and CUN patients showed a positive correlation between plasma level of IL-6 and the number of PD-L1 molecules (Fig. 6e). On the other hand, comparing IL-6 plasma level with HLA-DR revealed that COV patients, but not CUN patients, showed a negative correlation between these two parameters (Fig. 6e).

**COV patients are characterized by lung fibrosis and activated macrophages.** Given that COV patients are characterized by profound immune alterations detectable in the blood, we aimed in better characterizing the immune feature of lung micro-environment. We interrogated a public dataset containing Imaging Mass Cytometry (IMC) data performed on lung biopsies from patients who died because of severe SARS-CoV-2 infection, and we stratified patients as CUN and COV patients[40]. In the original paper, subjects were divided into two groups ('early' and 'late'), according to the timing of death (before or after 30 days from the start of respiratory symptoms, respectively). Increased level of infiltrating macrophages and fibrosis score in 'late' SARS-CoV-2 subjects have been displayed.

Re-analyzing the data stratified as CUN and COV, we observed that COV had higher deposition of collagen type I if compared to CUN patients (Fig. 7a, b). An increased deposition of collagen type I mediates lung fibrosis. Although, we did not observe any differences in terms of number of lung infiltrated macrophages, as outlined by CD68 expression (Fig. 7d), COV patients showed higher expression of vimentin positive macrophages. Vimentin is secreted by activated macrophages[41] and here co-localized with cleaved caspase-3 and alveolar epithelial cells expressing keratin 8 and keratin 18 (KRT8 + KRT18 +). These observations could indicate that the activation of tissue macrophages is higher in lungs from COV than in CUN patients (Fig. 7c, d, Supplementary Fig. 8).

**Discussion**
In this study, we describe some relevant immunological changes in aged patients with severe COVID-19 compared to younger ones, that can be briefly summarized as follows. Older patients display: 1) higher level of pro-inflammatory cytokines; 2) higher percentages of plasmablasts; 3) lower percentages of cTfh cells; 4) reduced plasmatic level of anti-S and anti-RBD IgG3 isotype during first week of infection; 5) presence of antigen-specific cTfh cells with a less activated transcriptomic profile; 6) lower percentages of CD4+ CM T cells with high expression of PD-1; 7) higher percentages of NK cells expressing CD57; 8) higher percentages of circulating immature monocytes and pro-inflammatory monocytes expressing CD56, characterized by a lower expression of HLA-DR molecules; 9) lung resident, activated macrophages that contribute to microenvironment remodeling, likely promoting collagen deposition and fibrosis.

When compared to CUN, COV patients showed higher plasma level of IL-6, lower plasma levels of IFN-γ and of growth factors

promoting wound healing. Many patients affected by COVID-19 develop a rapid hyper-immune reaction sustained by cytokines leading to alveolar infiltration by macrophages and monocytes[42]. It is well known that IL-6 is one of the main mediators of inflammatory and immune response initiated by infection, and increased levels of IL-6 are present in patients with COVID-19, and are associated with a strong inflammatory response, respiratory failure, necessity for mechanical ventilation and/or intubation, and mortality[43,44]. As a result, inhibiting the activity of cytokine by biological drugs is a successful therapeutic strategy[45,46]. Plasmatic IL-6 is normally extremely low - if not undetectable - in healthy adults, but elevated levels of IL-6 have been reported in older adults and are associated with disability and mortality[47]. Thus, we could hypothesize that in COV patients SARS-CoV-2 infection exacerbates all the phenomena that characterize the chronic state of subclinical inflammation defined as inflammaging, so contributing to increase the inflammation typical of the first phases of this infection.

We found that the percentage of memory B cells of aged patients was lower if compared to young patients, but the percentage of plasmablasts was higher. The specificities and affinities of plasmablasts have been reported for several viral infections, so far most extensively for influenza and HIV. In general, the immunoglobulin variable regions of plasmablasts are highly mutated and diverse, suggesting that plasmablasts are derived from memory B cells that were activated before differentiation[48]. Here, in COV patients the percentage of plasmablasts inversely correlates with the percentage of circulating Tfh cells, that are a specialized CD4+ T subset that provides signals to B cells, guides their development through the germinal center, and are necessary for class-switched antibody responses. cTfh cells have similarity to lymph node resident Tfh cells, have with potent B-cell helping ability and are mobilized into circulation after infection and vaccination. Overall, they generally reflect ongoing germinal center reactions[49,50]. Our data could suggest that plasmablasts expansion in COV patients, which is accompanied by a decrease of cTfh cells, appear in the circulation after the infection and it is partially independent from cTfh-plasmablast signalling. Therefore, during the early phases of acute SARS-CoV-2 responses, an uncoordinated B-T response in COV patients likely fails to control efficiently the disease. Indeed, we saw in COV lower attenuated IgG3 responses, accompanied by compromised Fcγ receptor binding and Fc effector activity, indicating the presence of an uncoordinated and delayed humoral activity rather than a fully dysfunctional humoral response. Finally, along with the reduction of cTfh cells, we also observed that SARS-CoV-2-reactive cTfh cells of COV patients expressed fewer genes involved in T cell activation; this observation could explain the consequent decrease in their functional ability to help B cells.

SARS-CoV-2 causes per se an activation of the innate and adaptive immune system, resulting in the depletion of the reservoir of naïve T cells and in a rapid redistribution of the memory subsets[51]. Similarly, aging is accompanied by a remodeling of the immune system with a reduction in T naïve cells and

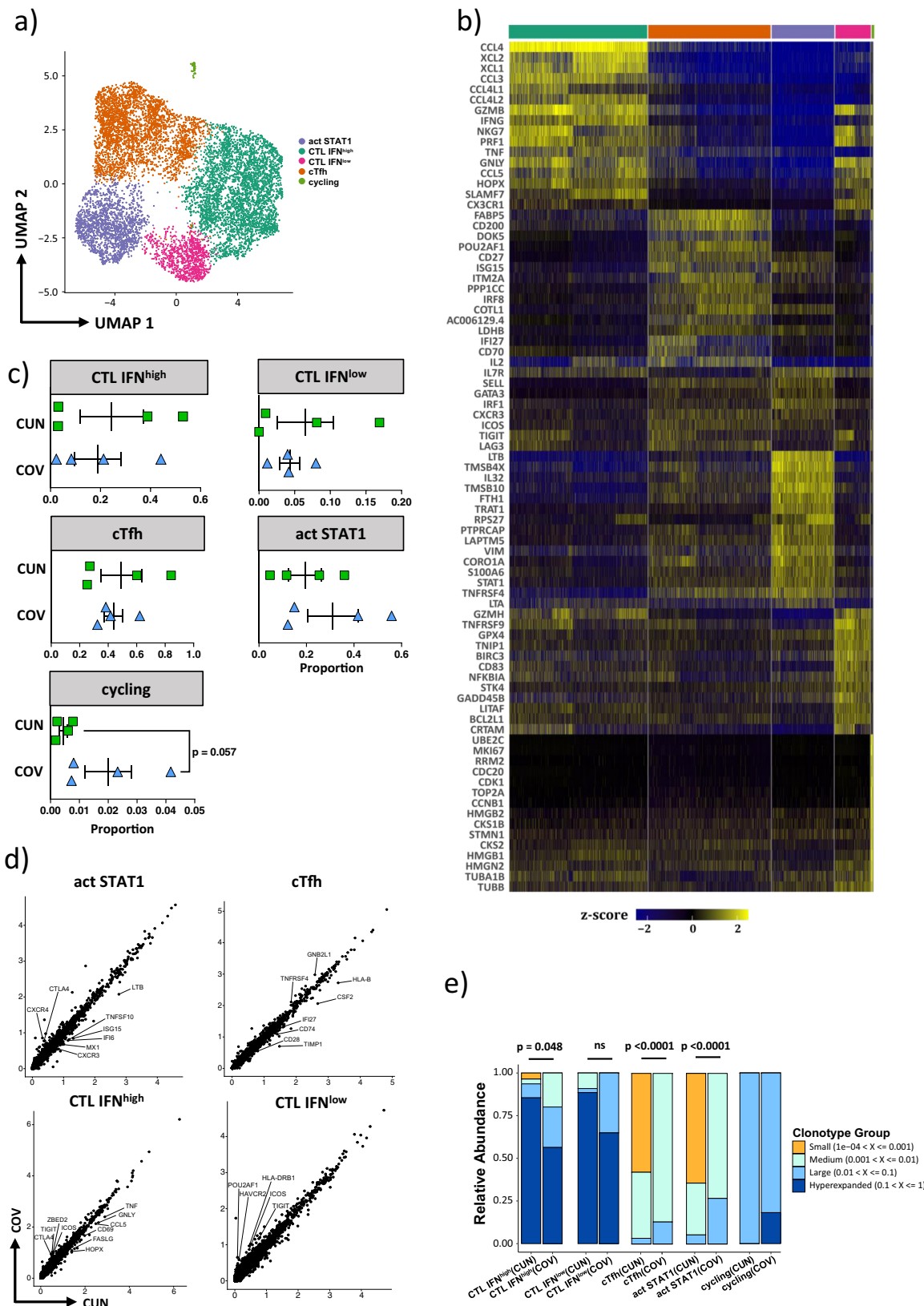

a relative increase in the frequency of memory T lymphocytes[52]. Previous data suggest that aging and scarcity of naïve T cells may be risk factors for failure to generate a coordinated adaptive immune response, resulting in increased susceptibility to COVID-19[53]. We found that COV patients display lower percentages of CM CD4+ T cells, but they express higher levels of

PD-1 on the cell surface, likely suggesting that in these patients a compensatory mechanism could exist that attempts to mount a better immune response.

Aged patients were characterized by higher percentage of CD57 + NK cells, a particular population of highly cytotoxic elements at the final stage of their maturation. This population

**Fig. 5 Single cell transcriptomic analysis of CD4+CD154+CD69+ T cells. a** UMAP plot shows the distribution of 8959 SARS-CoV-2-specific cells from CUN (n = 4) and COV (n = 4) patients. **b** Heatmap displays scaled-expression values of discriminative gene set per cluster related to CD4+CD154+CD69+ T cells that passed quality control. A list of the most representative genes is shown for each cluster (left). The upper bar colors were matched with those reported in the UMAP. Activated STAT1+ (act STAT1); cytotoxic T cells (CTL); circulating T follicular helper (cTfh). **c** Dotplots show the proportion of each cluster among CUN and COV patients. The central bar represents the mean ± SEM. Mann-Whitney U-test. **d** Dotplots reporting the DEGs among CUN and COV samples within each cluster. No DEGs were reported between cycling cells of CUN and COV (threshold: FDR < 0.01 and log₂FC > 0.3). **e** Occupied repertoire space for each scRNA-seq cluster among CUN and COV patients. Changes in the frequencies of hyperexpanded (for CTL IFN^high and IFN^low) and rare (for cTfh and act STAT1) clones between CUN and COV were assessed using Mann-Whitney U-test; exact p-values are reported in the figure.

increases with age[54], is associated with infections[55], and has been described also in COVID-19 asymptomatic pregnant women[26]. The increased proportion of CD57 + NK cells in the elderly likely explains the maintenance of cell cytotoxic responses despite the reduction in the activity of T cells typical of the physiological or pathological process of immunological aging[56].

Acute inflammation is triggered by myeloid cells, including monocytes and macrophages, and plays a main role in the pathogenesis of severe COVID-19. Here, we found that COV patients exhibit: i) a higher percentage of PD-L1 + intermediate monocytes; ii) decreased HLA-DR expression in total monocytes; iii) a higher proportion of classical monocytes expressing CD56, referred to as dysregulated pro-inflammatory monocytes producing basal level of IL-6 without stimulation. The presence of higher levels of PD-1 and PD-L1 on monocytes from severe COVID-19 has been linked to metabolic dysfunctions, even if altered mitochondrial ultrastructure are unable to impair their ability to produce pro-inflammatory cytokines[13]. Other than ultrastructure, cytomorphologic changes have been described, and indeed the monocyte distribution width (MDW) can now be considered a novel cytometry-based marker of disease severity with a relevant prognostic significance[57]. If several evidences confirm that a hyperinflammatory signature characterizes monocytes from COVID-19 patients, other data prove that an immunosuppressive signature, defined by downregulation of HLA-DRA, ALOX5AP, and RETN, also exists. Interestingly, the former is shared with HIV infection, the latter with sepsis[58]. Monocytes and macrophages accumulate in lung of severe COVID-19 patients, exhibit hyperinflammatory signatures, produce chemokines, such as CCL2 and CCL3; their frequency is correlated with older age and mortality[59]. Moreover, a marked increase in CD163+ myeloid cells aggregated in the alveolar spaces, which are crucial sites for oxygenation, has been observed in lung autopsy specimen[59].

Finally, the fine analysis of in silico data that we report here reveals that in the lung of COV patients both an increased deposition of collagen type I and significant macrophages activation exist that likely contribute to the development of inflammatory and fibrotic sequelae. This is in agreement with recent radiological findings showing that lungs of aged COVID-19 convalescents display heterogeneous imaging features suggestive of ongoing inflammation and fibrosis[60]. This could suggest that the immune traits here reported could be, at least in part, the cause of post-COVID lung sequelae in older individuals.

We are well aware that this study has several limitations. Given the paucity of biological material that we could obtain from each patient, we could not perform functional experiments in all of them. Moreover, because of the lockdown and the restriction in mobility, we could not recruit a larger group of healthy donors aged >70 years. However, this is the one of the first studies that broadly evaluates the immune traits in severe COVID-19 > 70 years old, describing molecular and cellular features of several immune cell types that undergo relevant changes in normal aging. Thus, the basal proinflammatory status typical of inflammaging coupled with potent inflammation triggered by SARS-CoV-2 infection can lead to a progressive inability to mount an adequate immune response. Without considering the highly protective and beneficial effects of vaccination – the molecular and cellular changes that we have described, intrinsically due to aging per se, are unavoidable major risk factors for developing severe and critical SARS-CoV-2 infection.

## Methods

**Patients**. Sixty-four COVID-19 patients with severe symptoms were enrolled in the study. Patients were admitted to the Infectious Diseases Clinics or to the Intensive Care Unit (ICU) of the Azienda Ospedaliero Universitaria (University Hospital) in Modena; all patients had an acute infection and a positive SARS-CoV-2 nasal swab test by PCR. Each participant, including healthy donors, provided informed consent according to Helsinki Declaration, and all uses of human material have been approved by the local Ethical Committee (Comitato Etico dell'Area Vasta Emilia Nord, protocol number 177/2020, March 11th, 2020 and subsequent amendments) and by the University Hospital Committee (Direzione Sanitaria dell'Azienda Ospedaliero-Universitaria di Modena, protocol number 7531, March 11th, 2020). Plasma samples and cells from a total of 32 adult healthy donors (HD, mean age of 54.1 ± 15.9) with no prior diagnosis or recent symptoms consistent with SARS-CoV-2 illness and with a negative serology were used for some analyses.

**Blood processing**. Peripheral blood was collected immediately after admission, before starting any therapy, put into ethylenediaminetetraacetic acid (EDTA) tubes and immediately processed. Isolation of peripheral blood mononuclear cells (PBMC) was performed using ficoll-hypaque according to standard procedures[61]. PBMC were rapidly stored in liquid nitrogen in fetal bovine serum (FBS) supplemented with 10% dimethyl sulfoxide (DMSO). The plasma was stored at −80 °C. Measurements were taken from individual patients; in the case of plasma, each measurement was performed in duplicate and only the mean was considered and shown.

**Detection and quantification of cytokines in human plasma**. Plasma levels of 62 cytokines were measured in 23 COV and in 26 CUN, as well as in 10 healthy donors with a mean age of 43.1 ± 12.6, using a Luminex platform (Human Cytokine Discovery, R&D System, Minneapolis, MN). The panel included BMP-7, BMP-2, BMP-4, IL-11, TACI, Fas, FasL, Leptin, Leptin-R, APRIL, BAFF, IL-18, IL-23, IL-6Ra, OPN, IL17C, IL-19, IL-27, CCL2, CCL3, CCL4, CCL5, CCL19, CCL11, CCL20, CXCL1, CXCL2, CXCL10, CX3CL1, CD40L, FGF-β, G-CSF, GZMB, IL-1α, IL-1β, IL-1ra, IL-3, IL-2, IL-4, IL-5, IL-6, IL-7, IL-12p70, IL-13, IL-10, IL-15, IL-17, IL-17E, IL-17C, IL-33, PDGF-AA, PDGF-AB/BB, TGF-α, TRAIL, EGF, FLT-3 ligand, GM-CSF, IFN-α, IFN-β, IFN-γ, TNF, VEGF, PD-L1. Each dot represents the mean of two technical replicates. The dashed line represents the median value of the cytokine within the HD group.

**Mass cytometry**. Thawed PBMC were washed twice with PBS and stained with Maxpar® Direct™ Immune Proling Assay™ (Fluidigm), a dry 30-marker antibody panel (viability marker Cell-ID™ Intercalator-103Rh included) plus the addition of 6 drop-in catalog antibodies (Fluidigm) and 2 custom-conjugated mAbs, for a total of 38 markers. The markers were the following: CD3, CD19, CD45, CD4, CD20, CD45RA, CD8, CD25, CD45RO, CD11c, CD27, CD56, CD14, CD28, CD57, CD16, CD38, CD66b, CCR7, CXCR3, CXCR5, HLA-DR, IgD, TCRγδ, CD123, CD127, CD161, CD294, CCR4, CCR6, CXCR1, PDL1, CD80, CD40, CD24, PD1-1, CD11b/MAC, CD21, IgM. See Supplementary Table 1 for the complete list of mAbs used. At least 300,000 events were acquired per sample. Data in FCS format were normalized for intra-file and inter-file signal drift using the FCS Processing tab in the CyTOF Software 6.7. The method is a two-step algorithm that first identifies the EQ Four Element Calibration Beads and then applies the dual count values registered by the beads to calculate the normalization factor to be applied to the data.

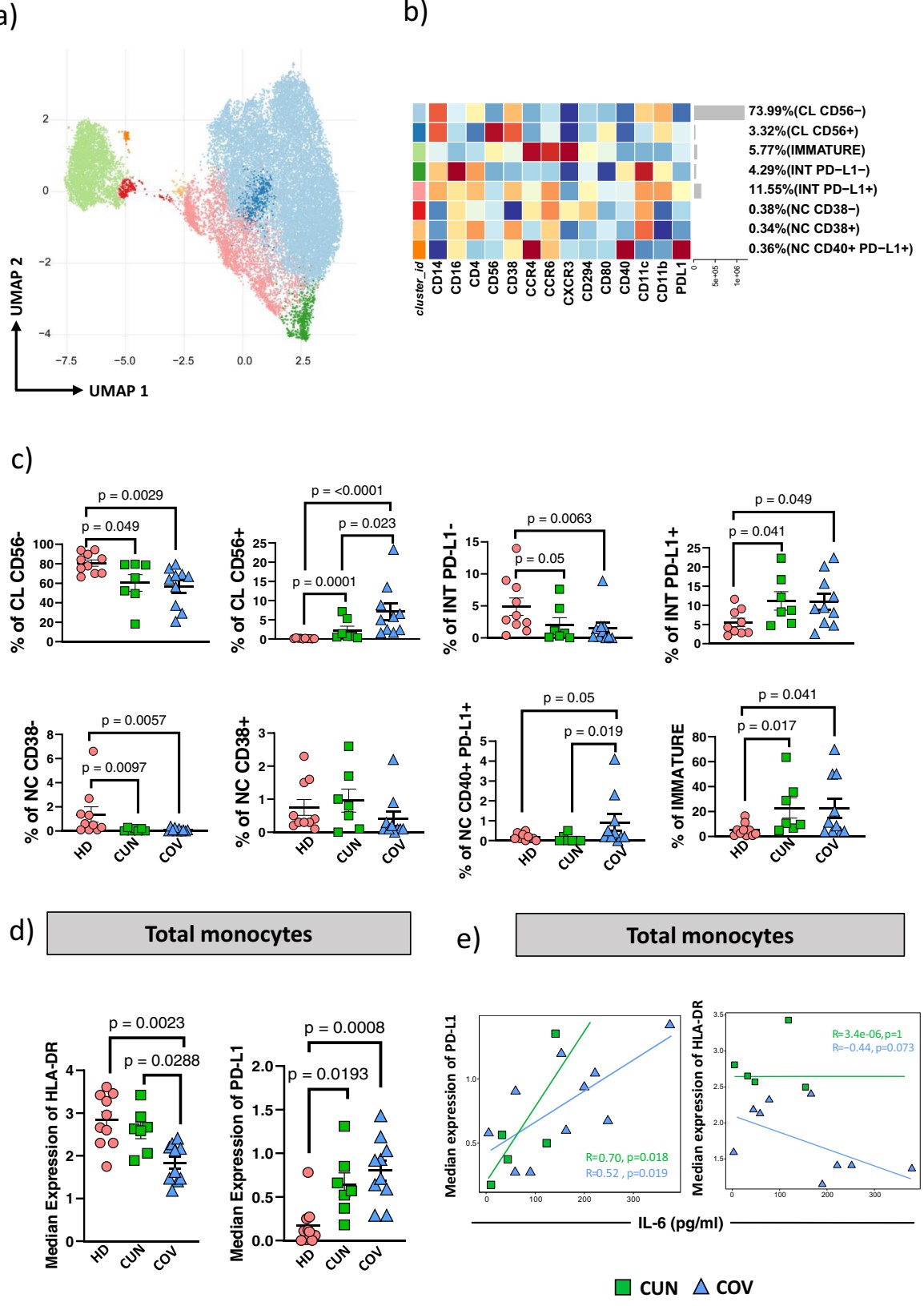

**Proliferation assay**. PBMC were stimulated for 6 days in resting conditions (as a control) or after stimulation with anti-CD3 plus anti-CD28 mAbs (1 μg/mL each, Miltenyi Biotech, Bergisch Gladbach, Germany) and with 20 ng/mL IL-2. The fluorescent dye 5,6-carboxyfluorescein diacetate succinimidyl ester was used at a concentration of 1 μg/mL (ThermoFisher Scientific) according to standard procedures[61]. Flow cytometric analysis for the identification of cycling cells belonging to different lymphocyte populations was performed on CD19+ B cells

and on electronically gated population of $T_N$, $T_{CM}$, $T_{EM}$, $T_{EMRA}$ CD4+ or CD8+ T cells. Supplementary Table 2 reports the complete list of mAbs used.

**Cell sorting of central memory (CM) CD4+ T lymphocytes**. Thawed PBMC were washed twice with RPMI 1640 supplemented with 10% fetal bovine serum

**Fig. 6 Monocytes reclustering in CUN and COV patients. a** UMAP plot shows the 2D spatial distribution of 1,608,200 cells from healthy subjects (HD, $n = 10$), CUN ($n = 7$) and COV ($n = 10$) embedded with FlowSOM clusters. **b** Monocytes heatmap shows the median marker intensities of the 14 lineage markers across the 8 cell populations obtained with FlowSOM after the manual metaclusters merging. Lightgrey bar along the rows (clusters) and values in brackets indicate the relative sizes of clusters. CL classical monocytes, INT intermediate monocytes, NC non-classical monocytes. **c** Dotplots show the relative cells percentage of the 8 clusters among healthy donors (HD; salmon circles; $n = 10$), CUN (green squares; $n = 7$) and COV (azure triangles; $n = 10$). The central bar represents the mean ± SEM. GLMM test was used for the statistical analysis. Exact $p$-values are reported in the figure. **d** Dotplots show the surface median expression (number of molecules) of the HLA-DR and PD-L1 on total monocytes among CUN and COV patients. Total monocytes were selected from CyTOF unsupervised analysis (Fig.2). The central bar represents the mean ± SEM. Kruskal-Wallis test with Benjamini-Hochberg correction for multiple comparisons was used to test the differences among the three groups; $p$-values are indicated in the figure. **e** Spearman correlation between plasmatic IL-6 and PD-L1 median expression on total monocytes in both CUN and COV patients (left); Spearman correlation between the plasmatic IL-6 and median expression of the HLA-DR on total monocytes in both CUN and COV patients (right).

and 1% each of l-glutamine, sodium pyruvate, nonessential amino acids, antibiotics, 0.1 M HEPES, 55 μM β-mercaptoethanol and 0.02 mg/ml DNAse.

CD2$^+$ cells were isolated through immunomagnetic separation technique (Miltenyi Biotec, Bergish Gladbach, Germany). Purified CD2$^+$ cells were washed with PBS and stained 20' at room temperature using LIVE DEAD Red (ThermoFisher) viability marker. Then, cells were washed and stained for 20' at room temperature with the following mAbs: anti-CD4 FITC, -CCR7 PE and -CD45RA PE-Cy7 (Biolegend, San Diego, CA, USA).

CM CD4$^+$ T cells were identified and sorted with a purity more than 98% by using a Biorad eS3 Sorter (Bio-Rad, Hercules, California, USA), operating inside a specifically designed Benchtop Biocontainment Enclosure (bioBUBBLE, Fort Collins, Colorado, USA). The gating strategy and purity are shown in Supplementary Fig. 9.

**RNA isolation, quantification and RNA sequencing**. Total RNA was extracted from sorted CM CD4$^+$ T cells by using Quick-RNA™ Miniprep Kit (Zymo Research, Irvine, CA, USA) and eluted in 30 μL of DNase/RNase-Free water. Therefore, RNA was quantified using RNA 6000 Nano Reagent (Agilent Technologies, Santa Clara, CA) on a 2100 Bioanalyzer system (Agilent Technologies, Santa Clara, CA). Only samples with RNA integrity number (RIN) ≥ 7 were selected to be sequenced. Libraries for total RNA sequencing were prepared from 300 ng total RNA using the Illumina Stranded mRNA Prep (Illumina) and analysed using NovaSeq 6000 Sequencing System.

**Representation of high-parameter mass cytometry**. Normalized CyTOF files were exported as.fsc, compensated using the CATALYST pre-processing workflow. Then all.fcs files were uploaded in FlowJo software v10.7.1 and checked to exclude aggregates, doublets, dead cells and non-biological events. For each sample, we therefore selected data from all living CD45$^+$ cells and imported them in R using flowCore package v2.4.0[62], for a total of 11,153,288 PBMCs. The further analysis was performed using CATALYST v1.14[63]. All data obtained by CyTOF were transformed using hyperbolic arcsin (asinh x/5). Clustering and dimensional reduction were performed respectively using FlowSOM and UMAP algorithms.

Concerning PBMC, we found 35 metaclusters, that were reduced to 23 by using manual merging of populations with similar characteristics. The CD4$^+$ and CD8$^+$ T cell clusters have been re-analyzed more in depth performing a new step of clustering and dimensional reduction using the following markers: CD45RA, CCR7, CD25, CD27, CD28, CD127, CD57, PD-1, CCR6, CCR4, CXCR3, CXCR5, HLA-DR, CD38 and CD294. Starting from 30 clusters of either CD4$^+$ T cells or CD8$^+$ T cells, this reclustering gave origin to 22 clusters of CD4$^+$ T cells and 19 of CD8$^+$ T lymphocytes.

Monocytes were re-analyzed using the following markers: CD14, CD16, CD4, CD56, CD38, CCR4, CCR6, CXCR3, CD294, CD80, CD40, CD11b, CD11c and PD-L1. Manual merging was performed to reduce the number of clusters from the initial 11 to 8 metaclusters. HLA-DR was not included in the monocyte clustering as all monocytes express HLA-DR, but at different level. This has been done to avoid issues during clustering due to subtle different level of HLA-DR expression.

The number of molecules expressed on the surface of T cells and monocytes such as PD-1, PD-L1 or HLA-DR was assessed by calculating the median expression of the marker using *plotExprHeatmap* function, a part of CATALYST package, and exporting the matrix values. These values were then analyzed as quantitative variables and compared using Mann–Whitney $U$-test, each $p$-value was reported in the figure. A quality control was performed for PBMCs, CD4$^+$ and CD8$^+$ T lymphocytes and monocytes to check inter-file signal drift to avoid shifts in clustering step between samples (technical and rather than biological variance) as reported in Supplementary Figs. 10–15.

**Flow cytometry analysis**. PBMC were washed with PBS and stained with the viability marker LIVE DEAD Aqua (Thermo Fisher Scientific). Then, up to 2 million cells were washed and stained at 37 °C with directly conjugated mAbs anti-CXCR5-APC and -CCR7-FITC. Cells were washed with PBS and stained at room temperature in Brilliant Stain Dye Buffer (BD Bioscience) with surface anti-CD3-

PB, -PD-1-BV605, -CD4-AF700, -CD8-APC-Cy7 and -CD45RA-PE-Cy7 mAbs (see Supplementary Table 3 for the complete list of the mAbs that were used). A minimum of 500,000 PBMC were acquired by using Attune NxT acoustic focusing flow cytometer (Thermo Fisher Scientific). Then all compensated.fcs file were uploaded in FlowJo software v10.7.1 and analyzed.

**Principal component analysis**. Principal Component Analysis (PCA) was performed and visualized in R using *prcomp* and *pca3d* package. To perform PCA we used a matrix containing the percentage of clusters for each sample based on the results of unsupervised data analysis performed with CATALYST 1.14. The total contribution of a given variable retained by PC1 and PC2 is equal to [(C1 * Eig1) + (C2 * Eig2)]/(Eig1 + Eig2), where C1 and C2 are the contributions of the variable on PC1 and PC2 Eig1 and Eig2 are the eigenvalues of PC1 and PC2.

**Bulk RNA-seq data analysis**. FASTQ files containing sequences data were quality-controlled using FastQC version 0.11.9. Each sample was paired-end sequenced at the depth of $40\times10^6$ reads using fragments length of 100 base pair (bp). Paired-end reads were aligned to the human genome (GENCODE Human Release 37, reference genome sequence GRCh38/hg38) using Rsubread v2.4.3[64]. Alignments were performed using default parameters. Reads associated with annotated genes were counted using the "featureCounts" function of Rsubread. Transcriptome analysis was performed using the edgeR package v3.32.1. Genes that have very low counts across all the libraries were removed prior to downstream analysis using HTSFilter package v1.32.0[65]. Library size normalization by trimmed mean of M values (TMM)[66] was performed by using the "calcNormFactors" function embedded in edgeR. Differential gene expression was assessed using "exactTest" function of egdeR, using default parameters. Benjamini–Hochberg correction was applied to estimate the false discovery rate (FDR). Differential expressed genes (DEG) were selected using as threshold FDR ≤ 0.05 and log$_2$FC > 0.5.

**In silico scRNA-seq and TCR-seq analysis**. Single-cell RNA-seq and TCR-seq data were retrieved from the Gene Expression Omnibus (GSE152522). The analysis was restricted to 8 hospitalized patients with severe COVID-19 pneumonia who were divided into two groups, i.e., CUN (age <60, $n = 4$; mean = 54.2 and sd = 3.8) and COV (age>70, $n = 4$; mean = 76.2 and sd = 4.3). The samples used were those identified by the codes: P01, P03, P06, P07, P12, P15, P17, P19. Pre-processed count matrix was imported in Rstudio using Seurat v4.0.1[67] and cleaned selecting the cells expressing <6% of mitochondrial genes and >60% of housekeeping genes. A total of 8378 cells were normalized using *SCTransform* function regressing for mitochondrial genes percentages. We performed clustering and dimensional reduction using UMAP finding five clusters at the resolution of 0.2. The resolution was selected using Clustree package v0.4.3[68]. Signature of each cluster was obtained by using "FindMarkers" function coded in the Seurat R package. Genes with a logFC > 0.3 and adjusted $p$-value <0.005 were selected as significant. The "FindMarkers" function was also used to identify the differential genes expression inside each cluster of CUN or COV patients. Mean expression of CUN and COV clusters were taken into account to generate the scatter plots. Finally, we combined the mRNA expression object with the TCR-seq data (cd4t6_annotation.txt) using *combine expression* function of scRepertoire v1.0.0 tool[69]. To examine the relative space occupied by clones at a specific proportion within each cluster among groups, we performed *clonalHomeostasis* function using as "cloneCall" parameter the CDR3 aminoacidic sequence. Furthermore, the frequency of different clones in each cluster for each patient was used to test the unique distribution rare clones between CUN and COV.

**Imaging mass cytometry (IMC)**. Publicly available IMC data were downloaded from https://doi.org/10.5281/zenodo.4110560 from the original paper[40]. For our purpose, to quantify the degree of lung fibrosis and macrophages infiltration, we selected three CUN patients (COVID_10-12-30; median age 55.1 ± 3.1 years old) and three COV patients (COVID_3-11-24; median age 72.0 ± 2.3 years old). For each selected patients all region of interest (ROIs) were analyzed. The images were first visualized using MDC Viewer v1.0.560.6 and then exported for secondary

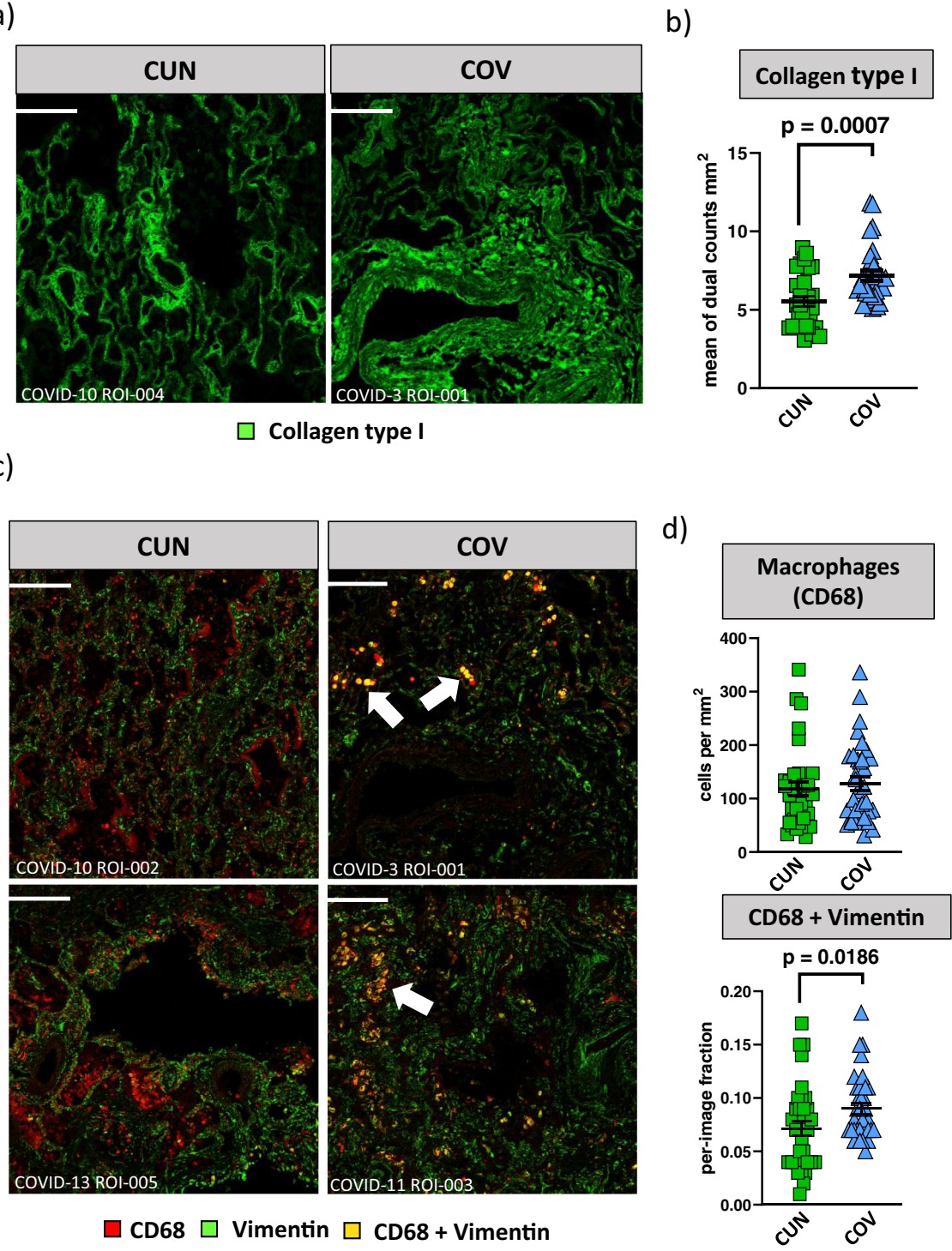

**Fig. 7 Spatial landscape of lung tissue from young and aged COVID-19 patients. a** Collagen type I distribution in lung images of CUN (*n* = 3; ROI = 31) and COV (*n* = 3; ROI = 30). Scale bar 200 µm. The sample name and the region of interest (ROI) are indicated. **b** Collagen type I mean intensity in images from CUN and COV patients. The central bar represents the mean ± SEM. Mann–Whitney *U*-test was used for the statistical analysis. Exact *p*-value is reported in the figure. **c** Spatial distribution of macrophages expressing CD68 (red dots) and activated macrophages simultaneously expressing CD68 and vimentin (yellow dots). Scale bar 200 µm. The sample name and the region of interest (ROI) are indicated. **d** Abundance of macrophages (top) and the fraction of activated macrophages (bottom) in lung images of CUN and COV patients. The central bar represents the mean ± SEM. Mann–Whitney *U*-test was used for the statistical analysis. Exact *p*-value is reported in the figure.

analysis performed by CellProfiler v4.2.1[70]. All pipeline used were found at (https://cellprofiler.org/examples) and adapted for our purposes.

**Statistics and reproducibility**. Differential cell populations abundances analysis was performed using generalized linear mixed model (GLMM) implemented

within diffcyt package[71] applying FDR cutoff = 0.05, each *p*-value was reported in the figure. Statistical analysis of cytokines, CFSE ad markers expression was performed using GraphPad Prism version 8. Quantitative variables were compared using non-parametric One-way Anova Kruskal-Wallis test, followed by Benjamini and Hochberg correction, unless stated otherwise in the figure legend. Each *p*-value

is reported in the figure. Significance was assigned at $p < 0.05$, unless stated otherwise in the figure legend.

**Reporting summary**. Further information on research design is available in the Nature Research Reporting Summary linked to this article.

## Data availability

Mass cytometry (CyTOF) data files are available in the flowrepository.org database under accession code FR-FCM-Z48K except for three files (CTR_1_LG.fcs, CTR_2_RB.fcs, CTR_7_FA.fcs) that are in the folder FR-FCM-Z3GH. The raw data generated in this study are provided in the Source Data file. Transcriptomic data (RNA-seq) are available in the Gene Expression Omnibus (GEO) public repository under the accession number GSE181005.

## Code availability

CATALYST script, used for CYTOF analysis, is available at: https://github.com/HelenaLC/CATALYST).

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

## Acknowledgements
We are grateful to Dr. Federico Banchelli (0000-0003-0499-2505) - AOU Policlinico di Modena and University of Modena and Reggio Emilia - for statistical advice. S.D.B. and L.G. are Marylou Ingram Scholar of the International Society for Advancement of Cytometry (ISAC) for the period 2015–2020 and 2020–2025, respectively. We gratefully acknowledge Fluidigm Corporation (San Francisco, CA) for generous and unconditional support. Drs. Paola Paglia (ThermoFisher Scientific, Monza, Italy), Leonardo Beretta (Beckman Coulter, Milan, Italy), Luca Cicchetti (Labospace, Milan, Italy) and Paolo Santino (Fluidigm Co., San Francisco, CA) are acknowledged for their support in providing reagents and materials in this study, and for precious technical suggestions. This study was partially supported by "Bando ricerca COVID-19 grant number COVID-2020-12371808" from Ministero della Salute to AC and by unrestricted donations from Glem Gas spa (San Cesario, Modena, Italy), Sanfelice 1893 Banca Popolare (San Felice sul Panaro, Modena, Italy), Rotary Club Distretto 2072 (Clubs: Modena, Modena L.A. Muratori, Carpi, Sassuolo, and Castelvetro di Modena), C.O.F.I.M. spa & Gianni Gibellini, Franco Appari, Andrea Lucchi, Federica Vagnarelli, Biogas Europa Service & Massimo Faccia, Pierangelo Bertoli Fans Club and Alberto Bertoli, Maria Santoro, Valentina Spezzani and BPER Banca. Finally, special thanks to the patients who donated blood to participate in this study.

## Author contributions
D.L.T., S.D.B., A.N., L.G., A.P., A.Q., C.P., G.A., L.F., S.D., D.L., J.N., R.B., A.I., and M.Ma. carried out experiments; D.L.T. drafted the figures; D.L.T. and A.N. drafted and revised the tables; M.Me., M.G., E.F., G.G., S.B., C.M., A.V.M., M.N., T.T. followed patients; L.Go. drafted and revised the clinical tables; D.L.T. performed bioinformatic and statistical analyses; D.L.T., S.D.B., L.G., and A.C. conceived the study; S.D.B. and A.C. wrote the manuscript. All authors read and approved the paper.

## Competing interests
AQ, CP, GA, SD, DL, and JN are employers of Fluidigm Corporation. All other authors declare no competing interests.
