## [Peer Review File · Communications Biology]

Reviewers' comments:

Reviewer #1 (Remarks to the Author):

In the manuscript by Lo Tartaro et al the authors performed an in-depth immunological characterisation of COVID patients aged <60 years and >70 years old. The authors find differential profiles of soluble mediators in plasma, differential PBMC profiles by CyTOF and RNAseq of CD4 T cells. The authors provide further evidence of immunological differences between younger and older COVID-19 patients with severe disease, including increased fibrosis of the lung. The manuscript is rather dense but overall well written. The findings are overall interesting and relevant to the field. The following comments should be addressed prior to publication, including reconsideration of some of the statements made in the manuscript.

1. There is no information regarding the timing of samples post disease onset. Are these acute timepoints? Is this comparable for the COV and CUN groups?
2. It would be beneficial if the authors clarified the new insights obtained from their re-analysis of existing datasets compared to the original publications (in results or discussion).
3. Many of the cytokine/chemokines/etc (fig1) that reach statistical significance for the COV vs COV comparison, do not seem different from the HDs. Have the authors compared the COV and CUN groups with the HDs?
4. Can the authors provide a reference for the definition of the 'exhausted B cell population'?
5. It is unclear how the data presented in Sup Fig5 relate to the CD4 T cell phenotypes seen in the author's own dataset. Please consider revising that section.
6. Line 318-319: "This observation suggests that CUN patients may develop a broader antibody response against several immunogenic regions of the SARS-CoV-2 virus.". This seems like an overstatement as there is not measurement of antibody breadth, only TCR clonotypes, please revise.
7. Line 399-400: "indicating that SARS-CoV-2 sculpts the immune system inducing a likely ineffective memory B-cell response.". It is unclear why the authors think that a higher plasmablast response equals an ineffective memory B cell response. This seems like an overstatement, please revise.
8. Line 87: 'number of studied' should be 'number of studies'

Reviewer #2 (Remarks to the Author):

Aging is the strongest risk factor of severe covid and differences in immune responses by age critically play the roles. Lo Tartaro et al. conducted immune profiling study in severe covid patients with various modalities, analyzing serum cytokine levels, antibody levels, immune cell composition with CyTOF, RNA-seq of Tcm, lung tissue histochemical analyses, and examined the differences between young and old patients. They show higher inflammatory cytokine levels, lower SARS-CoV-2 antibodies coupled with lower cTfh, fewer Tcm and their transcriptomic difference such as higher coinhibitory/exhaustion signature, and lung fibrosis with macrophage activation in older patients. The study is pretty much extensive and the amount of data is impressive. However, a big issue throughout the paper is the lack of consistency in the presentation and handling of the control group, which makes the interpretation of their findings challenging. Ideally speaking, there should be age-matched control for young patient group and old patient group (young control and old control), throughout the study, as presented in Fig 3. While the difficulty in recruiting old controls as stated in Discussion is understandable, the inconsistency about the control group is quite concerning - in some figures, control data are shown only with the horizontal bar representing the mean values (Fig 1a-d, 2d, 3c), in some other figures, control data are shown with plots along with patients (Fig 4c, 6c-d), and in some other figure, the control group is plotted but seemingly

not included in the multiple comparison (Fig 2c), without any coherent explanation on why they are presenting the data and conducting the statistical comparison in that manner in each figure.

Major comments:

1-a) Fig.1. Only with the mean control values shown as the horizontal lines, we cannot at all tell if the observed differences are simply the effect of aging, or, due to the interactions between age and covid infection. For example, observed differences in IL-1a and IL-1b, are these small differences simply due to background age difference and something observed between healthy young and old individuals as well? It has been reported that aging is associated with IL-2 decline, but lower IL-2 in COV shown here simply represents its decline by the age?

As is briefly discussed in Discussion section as a limitation, I do understand the difficulties in recruiting especially old healthy controls. However, it is not adequate to reduce the control group data to just the horizontal lines of average values. Can authors subgroup healthy donors and show dot plots of young HD and old HD in parallel with CUN and COV? Even if it's difficult for small control number, as mentioned above, at least the presentation and statistical comparisons between groups including HD controls should be consistent and coherent throughout the paper.

1-b) line 144-146: "The hyper-inflamed status for both CUN and COV was also highlighted by the high levels of soluble GM-CSF, TNF, granzyme B and IL-18 as compared to healthy donors (HD)" - authors should present statistical testing that supports this statement. Elevation compared to the mean value of HD itself alone does not support this statement.

1-c) Figure 2c, legend says "GLMM test was used for comparison between COV and CUN" and it seems from the panels that for some reason HD group was excluded from comparison.

1-d) Figure 2d, 3c. Again, similar issue with Figure 1. Why control data are represented only by mean value dotted line and individual values not presented?

2) Figure 2e. Old HD and young HD could be colored in different colors so that how age itself has an impact on this PCA plot.

3) Since RNA-seq in Figure 3 is conducted with very small number of samples, $n = 3$ for each group, authors should increase N_s , or alternatively, at least some of the major genes reported here should be validated with qPCR to support their conclusion. Additionally, Venn diagram of DEG between 4 groups would be helpful.

4) It seems there is no description in the text, except for the data related to antibody titer assessment (Supplementary Fig 5), on what timing in the disease course the samples were collected. The cytokine levels and immune cell compositions dynamically change over time during the course of infection. The comparison would not be valid if these conditions are not comparable between the groups, and assessment and description on this point for each assay is necessary.

Minor comments:

1) Why many COV patients seems to have exactly the same IFN- β plasma levels? Limit of detection in one of the measurement batches? If so, how batch effects were taken care of between measurements for IFN- β and other cytokines?

2) Sex composition between COV and CUN is considerably different. Male 67% vs 90%. Is this statistically different (Fisher exact test of frequency?) Sex greatly affects immune responses. If different, this should be stated somewhere.

3) Figure 5d is interesting, but the cutoff was set to $\log_2FC = 0.3$, which roughly corresponds to just 20% of increase. Are these small increases functionally meaningful? Again not sure some of these weak differences can be explainable solely by the age difference.

4) Active caspase 3 in lung. Line higher expression of vimentin colocalized with active caspase 3. It is not clear what arrows ("apoptotic sites") in Supplementary Figure 8 are trying to show, nor

any description on it. I assume the authors are trying to show activated macrophages are damaging and causing apoptosis – in what kind of cells?

Reviewers' comments:

Reviewer #1 (Remarks to the Author):

In the manuscript by Lo Tartaro et al the authors performed an in-depth immunological characterization of COVID patients aged <60 years and >70 years old. The authors find differential profiles of soluble mediators in plasma, differential PBMC profiles by CyTOF and RNAseq of CD4 T cells. The authors provide further evidence of immunological differences between younger and older COVID-19 patients with severe disease, including increased fibrosis of the lung. The manuscript is rather dense but overall, well written. The findings are overall interesting and relevant to the field. The following comments should be addressed prior to publication, including reconsideration of some of the statements made in the manuscript.

1. There is no information regarding the timing of samples post disease onset. Are these acute timepoints? Is this comparable for the COV and CUN groups?

We thank the referee for this comment. Blood of COV and CUN patients has been taken at the hospital admission, so they are all acute timepoints and out of therapy.

2. It would be beneficial if the authors clarified the new insights obtained from their re-analysis of existing datasets compared to the original publications (in results or discussion).

We agree with the comment, and in the paper, we have evidenced this aspect in two separate points:

“In the original work, Meckiff et al showed that SARS-CoV-2-reactive CD4⁺ T cells from hospitalized patients compared to non-hospitalized patients had increased cytotoxic follicular helper and classical T helper cells, but lower level of regulatory T cells. Although hospitalized patients exhibited different age (range 33-82 years), no differences were reported in terms of clusters percentage, gene expression and TCR clonality”.

And

“In the original paper, subjects were divided into two groups (‘early’ and ‘late’), according to the timing of death (before or after 30 days from the start of respiratory symptoms, respectively). Increased level of infiltrating macrophages and fibrosis score in ‘late’ SARS-CoV-2 subjects have been displayed.

Re-analyzing the data stratified as CUN and COV, we observed that COV had higher deposition of collagen type I if compared to CUN patients (Figure 7a-b). An increased deposition of collagen type I mediates lung fibrosis. Although, we did not observe any differences in terms of number of lung infiltrated macrophages, as outlined by CD68 expression (Figure 7d), COV patients showed higher expression of vimentin positive macrophages. Vimentin is secreted by activated macrophages and here co-localized with cleaved caspase-3 and alveolar epithelial cells expressing keratin 8 and keratin 18 (KRT8⁺ KRT18⁺). These observations could indicate that the activation of tissue macrophages is higher in lungs from COV than in CUN patients (Figure 7c-d, Supplementary Figure 8).”

3. Many of the cytokine/chemokines/etc (fig1) that reach statistical significance for the COV vs COV comparison, do not seem different from the HDs. Have the authors compared the COV and CUN groups with the HDs?

Figures have been revised according to this comment, and the comparisons among groups are now now indicated, so that HDs were compared with COV and CUN, and p values are reported in the figures.

4. Can the authors provide a reference for the definition of the ‘exhausted B cell population’?

In our paper we had used this definition according to the suggestions of Moir S. (Evidence for HIV-associated B cell exhaustion in a dysfunctional memory B cell compartment in HIV-infected viremic individuals. *J Exp Med.* 2008 Aug 4;205(8):1797-805. doi: 10.1084/jem.20072683). However, as suggested by this comment, we prefer to change that definition with “atypical” B cells, given that this population is CD19⁺CD20⁺CD40⁺HLA-DR⁺CCR6^{low}CD1c^{low}CD27⁻CD21⁻CD24⁻CD38⁻IgD⁻IgM⁻ (As described here: Sutton H. et al., *Cell Reports* 2021, doi.org/10.1016/j.celrep.2020.108684).

5. It is unclear how the data presented in Sup Fig5 relate to the CD4 T cell phenotypes seen in the author’s own dataset. Please consider revising that section.

As suggested, we rephrase the sentence. ‘To explore more in depth this correlation...’ has been replaced with ‘To explore more in depth the correlation between cTfh and plasmablasts...’.

6. Line 318-319: “This observation suggests that CUN patients may develop a broader antibody response against several immunogenic regions of the SARS-CoV-2 virus.”. This seems like an overstatement as there is not measurement of antibody breadth, only TCR clonotypes, please revise.

The reviewer is right. We removed this sentence.

7. Line 399-400: “indicating that SARS-CoV-2 sculpts the immune system inducing a likely ineffective memory B-cell response.”. It is unclear why the authors think that a higher plasmablast response equals an ineffective memory B cell response. This seems like an overstatement, please revise.

We amended the manuscript as suggested, and we removed the sentence.

8. Line 87: ‘number of studied’ should be ‘number of studies’

We thank the reviewer for the note. We corrected the typo.

Reviewer #2 (Remarks to the Author):

Aging is the strongest risk factor of severe covid and differences in immune responses by age critically play the roles. Lo Tartaro et al. conducted immune profiling study in severe covid patients with various modalities, analyzing serum cytokine levels, antibody levels, immune cell composition with CyTOF, RNA-seq of Tcm, lung tissue histochemical analyses, and examined the differences between young and old patients. They show higher inflammatory cytokine levels, lower SARS-CoV-2 antibodies coupled with lower cTfh, fewer Tcm and their transcriptomic difference such as

higher coinhibitory/exhaustion signature, and lung fibrosis with macrophage activation in older patients.

The study is pretty much extensive, and the amount of data is impressive. However, a big issue throughout the paper is the lack of consistency in the presentation and handling of the control group, which makes the interpretation of their findings challenging. Ideally speaking, there should be age-matched control for young patient group and old patient group (young control and old control), throughout the study, as presented in Fig 3. While the difficulty in recruiting old controls as stated in Discussion is understandable, the inconsistency about the control group is quite concerning - in some figures, control data are shown only with the horizontal bar representing the mean values (Fig 1a-d, 2d, 3c), in some other figures, control data are shown with plots along with patients (Fig 4c, 6c-d), and in some other figure, the control group is plotted but seemingly not included in the multiple comparison (Fig 2c), without any coherent explanation on why they are presenting the data and conducting the statistical comparison in that manner in each figure.

Major comments:

1-a) Fig.1. Only with the mean control values shown as the horizontal lines, we cannot at all tell if the observed differences are simply the effect of aging, or, due to the interactions between age and covid infection. For example, observed differences in IL-1a and IL-1b, are these small differences simply due to background age difference and something observed between healthy young and old individuals as well? It has been reported that aging is associated with IL-2 decline, but lower IL-2 in COV shown here simply represents its decline by the age?

We observed a higher level of plasmatic level of IL-1 α , IL-1 β and IL-2 in CUN if compared to HDs and, as already described in literature, the infection induces an increase in the level of these pro-inflammatory cytokines. This increase is not detectable in COV patients, suggesting a weakened monocyte and lymphocyte response, likely compatible with such infection in aged patients.

As is briefly discussed in Discussion section as a limitation, I do understand the difficulties in recruiting especially old healthy controls. However, it is not adequate to reduce the control group data to just the horizontal lines of average values. Can authors subgroup healthy donors and show dot plots of young HD and old HD in parallel with CUN and COV? Even if it's difficult for small control number, as mentioned above, at least the presentation and statistical comparisons between groups including HD controls should be consistent and coherent throughout the paper.

We thank the reviewer for this comment. We agree that indicating HD data only with horizontal lines of average values was not fully clear. For this reason, we modified all the data related to values of HD making the presentation and statistical comparisons among groups (including HD) consistent and coherent throughout the paper. We amended this in Fig.1 and in the supplementary figures. We could have old HD and young HD controls only for RNA-seq experiments. HD group used for the quantification of plasmatic cytokines and phenotypic analysis has a mean age of 43.1 \pm 12.6 and this is reported in the method section. For the reasons already reported in the discussion, *i.e.* the difficulties to enroll old healthy subjects in those phases of the pandemics, we focused on the comparisons between young and old patients with COVID-19, pointing out the

immunological differences due to the effect of infection on the immune system, rather than comparison with healthy subjects.

1-b) line 144-146: “The hyper-inflamed status for both CUN and COV was also highlighted by the high levels of soluble GM-CSF, TNF, granzyme B and IL-18 as compared to healthy donors (HD)” - authors should present statistical testing that supports this statement. Elevation compared to the mean value of HD itself alone does not support this statement.

We agree with the reviewer and we amended this part as requested. Now all the dot plots report individual values of HD group together with statistic (mean and SEM). All the exact significant p values are reported in the figure.

1-c) Figure 2c, legend says “GLMM test was used for comparison between COV and CUN” and it seems from the panels that for some reason HD group was excluded from comparison.

We revised the legend as GLMM was used for all comparisons.

1-d) Figure 2d, 3c. Again, similar issue with Figure 1. Why are control data represented only by mean value dotted line and individual values not presented?

We amended as requested indicating individual values together with the statistic.

2) Figure 2e. Old HD and young HD could be colored in different colors so that how age itself has an impact on this PCA plot.

We thank the reviewer for the comment. We could not amend this request as in these experiments we do not have two groups of HDs, but only one aged 43.1 ± 12.6 year.

3) Since RNA-seq in Figure 3 is conducted with very small number of samples, $n = 3$ for each group, authors should increase Ns, or alternatively, at least some of the major genes reported here should be validated with qPCR to support their conclusion. Additionally, Venn diagram of DEG between 4 groups would be helpful.

We thank the referee for this suggestion. We amended as requested by showing Venn diagram in the figure 3 and in the supplementary figures. Moreover, we show in this response the Venn diagram of DEG among all the groups. To note, HUN/CUN and HOV/COV share 18 DEGs while HUN/HOV and HOV/COV share 6 DEGs. Furthermore, CUN/COV and HUN/HOV share 1 DEG while CUN/COV and HOV/COV share 1 DEG. Finally, HUN/CUN and HUN/HOV share 2 DEGs. No DEGs are shared between CUN/COV and HUN/HOV. Here below, we also report the table containing the ENTREZID of shared DEGs among each group.

Names	total	elements
CUN/COV HUN/CUN	1	644172
CUN/COV HOV/COV	1	3902
HOV/COV HUN/CUN	18	313 120892 4210 10875 22918 348110 5292 8778 375387 5029 3429 4778 91662 5175 4610 124599 4332 1896
HUN/CUN HUN/HOV	2	1050 80139
HOV/COV HUN/HOV	6	115703 8408 1291 5074 10223 51752
CUN/COV	5	2615 1493 1846 2833 1234
HUN/CUN	80	23092 221472 6256 5351 1026 57476 55160 3119 1545 7378 4035 10538 27244 51311 1959 3007 25805 3040 10803 9884 3728 3008 474170 54507 80727 3039 374739 5864 30009 10216 6515 1438 26050 84206 7045 4261 151963 90007 55510 338442 83937 8870 10628 54407 3306 901 55959 3311 7532 84627 8324 441478 375690 148641 79155 644246 3006 9839 9903 6714 1439 11033 144195 24138 8313 8843 84106 4214 100188953 26509 3164 79865 3090 10326 10871 23654 972 64231 79901 9619
HOV/COV	94	5724 29990 100506115 9315 319101 28616 25797 400499 10219 22903 58499 150372 100505696 978 50632 84941 80119 6546 4122 79661 123041 92126 23508 64393 55205 3487 414208 4175 503569 84747 10319 146722 55252 84329 112574 220158 1298 91304 4602 9985 649946 79854 102465537 8476 9397 9792 2212 216 10602 324 4998 22898 353514 1441 644165 51310 101929623 10409 2050 100534593 4599 1277 11240 9088 867 126917 101060391 8237 219972 4069 51191 116362 55701 3603 2334 283149 64388 11027 122618 84830 79887 338657 101954267 10536 3570 105 89848 387644 140807 144423 105377819 1593 23612 104326052
HUN/HOV	5	28514 253143 1117 84971 28316

Unfortunately, we cannot further amend this point by validating differential expressed genes by qPCR as we used all the biological material isolated from these patients. Since the project is over, the ethical committee (EC) is not valid anymore and we cannot recruit other participants with the same clinical and demographic characteristics without restarting with a new application to the EC, which would take a lot of time.

4) It seems there is no description in the text, except for the data related to antibody titer

assessment (Supplementary Fig 5), on what timing in the disease course the samples were collected. The cytokine levels and immune cell compositions dynamically change over time during the course of infection. The comparison would not be valid if these conditions are not comparable between the groups, and assessment and description on this point for each assay is necessary.

Thanks for the comment. All the samples used in the plasmatic cytokine and phenotype analysis and meta-analysis are similar for the timing in the disease course. All the blood samples were collected in the acute phase, generally immediately after hospital admission.

For the data reported in figure 5, the following patients were used for the analysis: P01, P03, P06, P07, P12, P15, P17 and P19. Clinical characteristics of the patients are reported in the original supplementary tables 1A and 2A (*Meckiff et al., 2020, Cell 183, 1340–1353, doi.org/10.1016/j.cell.2020.10.001*).

For data reported in figure 7, all patients (both COV and CUN) died because of severe COVID-19, and they were all analyzed postmortem. Patients selected for the analysis were VCOCID_2, COVID_12, COVID_10, COVID_30, COVID_24, COVID_11 and their characteristics are reported in the supplementary material of the original paper (*Rendeiro et al., The spatial landscape of lung pathology during COVID-19 progression. Nature. 2021 May;593(7860):564-569. doi: 10.1038/s41586-021-03475-6.*)

For the neutralizing Ab analysis, reported in supplementary figure 5, we selected the following patients:

- HOV: 6, 17, 18, 23, 36, 37, 40, 41, 261, 265, 266;
- HUN: 15, 19, 27, 32, 33, 42,45,200,201, 202, 204, 205,206, 229, 264;
- CUN: 1,20, 22, 76, 92, 98, 99, 109, 113, 123, 136, 134, 138, 139, 141, 147, 159, 176, 178, 210, 217, 220, 227, 272, 273, 289, 300;
- COV: 52, 66, 77, 80, 90, 97, 116, 120, 158, 193, 246.

Clinical characteristics of the patients are reported in the supplementary material of the original paper (*Zohar T, et al. Compromised Humoral Functional Evolution Tracks with SARS-CoV-2 Mortality. Cell 183, 1508-1519 e1512 (2020).*)

Minor comments:

1) Why many COV patients seems to have exactly the same IFN- β plasma levels? Limit of detection in one of the measurement batches? If so, how batch effects were taken care of between measurements for IFN- β and other cytokines?

Thanks for the comment. The reviewer is correct, the same value indicates the limit of detection, but there were no batches effect as all samples have been analyzed in the same day using Luminex platform (Human Cytokine Discovery, R&D System, Minneapolis, MN) with the same internal control and standard curves.

2) Sex composition between COV and CUN is considerably different. Male 67% vs 90%. Is this statistically different (Fisher exact test of frequency?) Sex greatly affects immune responses. If different, this should be stated somewhere.

As reported in table 1, sex composition between COV and CUN is different (Fisher's exact test). We amended as suggested, and we added a sentence in the patients' section.

3) Figure 5d is interesting, but the cutoff was set to $\log_2FC = 0.3$, which roughly corresponds to just 20% of increase. Are these small increases functionally meaningful? Again not sure some of these weak differences can be explainable solely by the age difference.

Thanks for the comment. In the original paper, the threshold for the DEGs was set to $\log_2FC = 0.25$ (used to include weak gene expression changes and used to perform pathway enrichment analyses). Here, we use a more stringent cutoff ($\log_2FC = 0.3$) to reduce the false discovery rate (FDR) and select genes that are influenced by age in response to SARS-CoV-2 infection.

Although we identified small increases in gene expression, likely such changes are still important as they derived from the analysis of a small dataset of 8,378 age-matched SARS-CoV-2-reactive CD4+ T cells. Furthermore, to increase the robustness of DEGs results and to reduce the effect of confounding features, we selected the patients with similar comorbidities, clinical parameters and no bacterial or fungal co-infection.

4) Active caspase 3 in lung. Line higher expression of vimentin colocalized with active caspase 3. It is not clear what arrows (“apoptotic sites”) in Supplementary Figure 8 are trying to show, nor any description on it. I assume the authors are trying to show activated macrophages are damaging and causing apoptosis – in what kind of cells?

Thank you for this comment. We revised the supplementary figure 8, by displaying two magnifications of the area in which activated macrophages, identified by vimentin and CD68 co-expression, colocalize with apoptotic sites (cleaved caspase-3) and alveolar epithelial cells expressing keratin 8 and keratin 18 (KRT8+ KRT18+, cyan dots) in lung biopsies from COV.

Reviewers' comments:

Reviewer #1 (Remarks to the Author):

The authors have addressed my comments.

Reviewer #2 (Remarks to the Author):

In this revision, improvements were made responding to the concerns raised to basic points, such as presenting the HD group data in some figures in which these were missing. However, the concern on the statistical comparisons including HD control group has not been resolved. For example, in Figure 1, it is described that the comparisons are done by Mann-Whitney test, and this testing seemed to be conducted throughout the paper in similar comparisons. Mann-Whitney test is essentially testing for two-group comparison. ANOVA testing or Kruskal-Wallis testing would be more appropriate for multiple group comparison testing. In some supplementary figures (5 and 7), there are descriptions that Mann-Whitney testing were used with Bonferroni correction, which at least takes care of the issue of multiple testing. However, in Figure 1a) – d), 3c), 6d), in which legends say "Mann-Whitney U-testing was used, exact p-values are reported", it is not obvious if any consideration is taken for multiple testing.

If these p-values are indeed the "exact p-values" of Mann-Whitney test between two groups compared, the threshold for null hypothesis (alpha value) is $p = 0.05/3 = 0.0166$ with Bonferroni adjustment, and in that case, IL-6 (CUN vs COV), IL-2 (HD vs CUN), IL-13/33/CXCL10/PD-L1, FAS-L (CUN vs COV), PD-1 Tcm (Fig 3) do not pass the threshold, i.e., not significantly different between these groups. Some of the conclusions discussed in the Discussion section are based on the results of these multiple, repeated Mann-Whitney testing within 3 groups, HD/CUN/COV. These points should be clarified, and relevant descriptions in the text should be corrected as necessary.

Point by point response

Reviewers' comments:

Reviewer #1 (Remarks to the Author):

The authors have addressed my comments.

We thank the reviewer for his/her comments that contributed to improve the quality of the manuscript.

Reviewer #2 (Remarks to the Author):

In this revision, improvements were made responding to the concerns raised to basic points, such as presenting the HD group data in some figures in which these were missing. However, the concern on the statistical comparisons including HD control group has not been resolved. For example, in Figure 1, it is described that the comparisons are done by Mann-Whitney test, and this testing seemed to be conducted throughout the paper in similar comparisons. Mann-Whitney test is essentially testing for two-group comparison. ANOVA testing or Kruskal-Wallis testing would be more appropriate for multiple group comparison testing. In some supplementary figures (5 and 7), there are descriptions that Mann-Whitney testing were used with Bonferroni correction, which at least takes care of the issue of multiple testing. However, in Figure 1a) – d), 3c), 6d), in which legends say “Mann-Whitney U-testing was used, exact p-values are reported”, it is not obvious if any consideration is taken for multiple testing. If these p-values are indeed the “exact p-values” of Mann-Whitney test between two groups compared, the threshold for null hypothesis (alpha value) is $p = 0.05/3 = 0.0166$ with Bonferroni adjustment, and in that case, IL-6 (CUN vs COV), IL-2 (HD vs CUN), IL-13/33/CXCL10/PD-L1, FAS-L (CUN vs COV), PD-1 Tcm (Fig 3) do not pass the threshold, i.e., not significantly different between these groups. Some of the conclusions discussed in the Discussion section are based on the results of these multiple, repeated Mann-Whitney testing within 3 groups, HD/CUN/COV. These points should be clarified, and relevant descriptions in the text should be corrected as necessary.

We thank the reviewer for his/her time, and we apologize for the inadvertence we had during the first round of revision. Taking advantage of your suggestion, we consulted the medical statistician, Dr. Federico Banchelli (<https://orcid.org/0000-0003-0499-2505>), of the University Hospital and University of Modena and Reggio Emilia. He suggested that in our study the most suitable test to perform is the non-parametric one-way Anova (Kruskall-Wallis test) followed by False Discovery Rate (FDR) control (with the Benjamini and Hochberg correction). He also said, that, as you mention, Mann-Whitney followed by Bonferroni correction could also be usable, but it is far more conservative. For this reason, we revised the statistic related to results reported in Figure 1, Figure 3c, Supplementary figures 1, 5 and 7. All the new p values are reported in the figures. At your disposal, regarding the results reported in figure 1 and supplementary figure 1, we report here below a table with the comparison of the “old” p value and “new” p value obtained by Kruskal-Wallis and Benjamini Hochberg correction. Moreover, all the data reported in the manuscript are also reported in the supplementary tables available to all.

Table reporting the P values obtained with Mann-Whitney test (three comparisons, “old p values”) vs p values obtained after Kruskal-Wallis test followed by Benjamini Hochberg correction (“new p values”). The comparison highlighted in yellow are those who lose the statistical significance after the Benjamini and Hochberg correction. As you can see, the main results that we present and discuss in the paper do not change by using more adequate statistical analysis.

	OLD P VALUES			NEW P VALUES		
	HD vs COV	HD vs CUN	CON vs CUN	HD vs COV	HD vs CUN	CON vs CUN
IL-6	<0.0001	<0.0001	0.0440	<0.0001	0.0004	0.0365
IL-11	0.1764	0.0070	<0.0001	0.3623	0.0424	0.0002
IL-1a	0.7358	0.0016	0.0039	0.5675	0.0064	0.0064
IL-1b	0.5024	0.0281	0.0044	0.8413	0.0443	0.0311
IL-2	0.9472	0.0288	0.0164	0.8460	0.0563	0.0432
IFN-B	0.0946	0.8381	0.0046	0.1213	0.7036	0.0204
IFN-G	>0.9999	0.1707	0.0067	0.1427	0.7410	0.0364
IL-13	0.9004	0.2555	0.0491	0.9518	0.2652	0.1992
IL-33	0.6679	0.1240	0.0173	0.9680	0.1050	0.0481
CCL5	0.9233	0.3034	0.0116	0.2043	0.6655	0.0363
CXCL2	0.0501	0.4127	0.0091	0.0290	0.5978	0.0290
CX3CL1	<0.0001	0.0041	0.0085	0.0002	0.0215	0.0215
CXCL10	<0.0001	<0.0001	0.0424	<0.0001	0.0002	0.0958
PDGF-AA	0.9846	0.1447	0.0098	0.1349	0.8510	0.0425
PDGF- AB/BB	0.7727	0.2137	0.0067	0.6040	0.2160	0.0292
EGF	0.4994	0.0281	0.0055	0.6250	0.0240	0.0240
PD-L1	<0.0001	0.0014	0.0423	0.0001	0.0089	0.0667
FAS	0.0491	0.4753	0.0042	0.1400	0.5806	0.0099
FAS-L	<0.0001	<0.0001	0.0236	<0.0001	0.0007	0.0602
BAFF	0.0002	0.0002	0.8349	0.0008	0.0008	0.8577
CD40L	0.0626	0.0027	0.0653	0.0899	0.0062	0.0899
IL-18	<0.0001	<0.0001	0.4993	0.0002	0.0005	0.5417
IL-27	0.0001	0.0149	0.2623	0.0017	0.0155	0.2228
CCL2	<0.0001	<0.0001	0.8660	0.0004	0.0004	0.8800
CCL19	<0.0001	<0.0001	0.5445	<0.0001	0.0001	0.6631
FGF-B	0.0291	0.0117	0.7253	0.0536	0.0454	0.7049
IL-1RA	<0.0001	<0.0001	0.7999	0.0001	<0.0001	0.6924
GRANZYME B	0.0001	<0.0001	0.6986	0.0001	0.0001	0.8481
IL-3	0.0941	0.0036	0.1202	0.1195	0.0123	0.1195
G-CSF	<0.0001	<0.0001	0.9963	0.0003	0.0003	0.9843
OPN	<0.0001	<0.0001	0.7134	0.0003	0.0004	0.7058

TRAIL	<0.0001	0.0004	0.1139	<0.0001	0.0022	0.1398
GM-CSF	<0.0001	<0.0001	0.7245	<0.0001	<0.0001	0.8052
CCL20	0.0225	0.0209	0.9446	0.0365	0.0365	0.9504
CCL11	0.2245	0.0108	0.1352	0.1988	0.0391	0.1902
TNF	0.0011	0.0036	0.4835	0.0037	0.0084	0.5007
VEGF	<0.0001	<0.0001	0.9446	<0.0001	<0.0001	0.9590
IL-10	0.0004	0.0004	0.6022	0.0012	0.0018	0.6462